# SARS-CoV-2 infection dynamics in a MHCI-mismatched lung transplant recipient

Jonas Fuchs [1,21], Vivien Karl [2,3,21], Ina Hettich[4,21], Jaime Alvarado [4], Daniel Eckert[3], Lena Jaki [1], Ann-Kathrin Kohl[1], Anastasia Kremser[2,3], Anastasija Maks[1], Charlott Terschluse[4,5], Prerana Agarwal[6], Florian Emmerich[7], Sebastian Fähndrich[4], Annabelle Flügler[8], Daniel Hornuss [9], Johannes Kalbhenn[10], Nikolaus Kneidinger[11,12], Inga Lau [13], Achim Lother[8,14], Isabelle Moneke[15], David Schibilsky[16,17], Elisabeth Schygulla[18], Nils Venhoff [19], Gernot Zissel [4], Martin Czerny[16], Daniela Huzly[1], Georg Kochs [1], Christoph Neumann-Haefelin [3,20], Bernward Passlick[15], Daiana Stolz[4], Robert Thimme [3], Marcus Panning [1,22] ✉, Maike Hofmann [3,22] ✉ & Björn C. Frye [4,22] ✉

A 48-year-old patient underwent lung transplantation because of severe COVID-19, which aggravated his underlying interstitial lung disease, despite the presence of detectable SARS-CoV-2. Subsequently, the graft is re-infected early in the post-procedural phase, leading to viral persistence for more than five months. By analyzing viral evolution and effector immune response within the transplanted organ, we observe three main findings. First, virus evolution differs in the transplanted organ compared to that in the upper respiratory tract and is affected by monoclonal SARS-CoV-2-specific antibodies and molnupiravir. Second, we show the potential clinical relevance of T cell HLA restriction that may facilitate viral clearance in the upper respiratory tract compared to the ongoing viral replication in the HLA mismatch organ. Third, close monitoring and modulation of immunosuppressive and antiviral therapy enables viral clearance in a lung transplantation setting despite incomplete SARS-CoV-2 clearance prior to transplantation.

Acute respiratory distress syndrome (ARDS) is characterized by severe hypoxic respiratory failure and bilateral pulmonary infiltrates[1]. Its etiology is heterogeneous and includes a wide variety of respiratory infections. Even with optimal clinical management, including extracorporeal membrane oxygenation (ECMO), ARDS-related in-hospital mortality ranges between 25% and 46% of affected persons, depending on severity[2–5]. In recent years, SARS-CoV-2 caused ARDS in thousands of people with high mortality. Current guidelines approve lung transplantation (LuTX) as a therapeutic option for select patients with SARS-CoV-2-related end-stage lung disease[6]. However, sufficient time without pulmonary recovery and complete clearance of the viral pathogen

is recommended[7,8]. Moreover, these guidelines are mainly based on prior experiences with otherwise healthy SARS-CoV-2 patients who have the potential for lung recovery even after severe COVID-19. In patients with preexisting lung diseases who are already listed for lung transplantation before SARS-CoV-2 infection, lung recovery after SARS-CoV-2 infection is less likely. Prolonged intensive care and ECMO therapy may hamper the success of lung transplantation[9,10]. Nevertheless, transmission of SARS-CoV-2 from the recipient to the donor organ is of concern, especially because persistent infection and acquisition of in-host mutations have both been described in organ-transplant patients[11,12]. These mutations could facilitate host-specific

A full list of affiliations appears at the end of the paper. ✉e-mail: marcus.panning@uniklinik-freiburg.de; maike.hofmann@uniklinik-freiburg.de; bjoern.christian.frye@uniklinik-freiburg.de

fitness advantages such as the selection of therapy-resistant quasispecies[13–15]. To date, transmission of SARS-CoV-2 from the recipient to the lung allograft has not been described.

Here, we show a case of a lung transplant recipient who became infected because of incomplete viral clearance prior to transplantation. After lung transplantation, SARS-CoV-2 divergently evolved in the upper (URT) and lower respiratory tract (LRT). Intriguingly, the virus was cleared in the URT but persisted in the LRT. Analyses of the CD8+ T cell response revealed inherent blindness of the recipient's emerging SARS-CoV-2-specific immunity towards the HLA-mismatched transplanted lung. Hence, we showed the clinical relevance of the immunological paradigm of HLA restriction in viral evolution and clearance. These findings have implications for careful monitoring and clinical management of virus-infected lung transplantation patients.

## Results

### Lung transplantation despite non-complete viral clearance

The patient is a 48-year-old Caucasian male who had systemic scleroderma with interstitial lung disease (SSc-ILD) for several years. Despite intensive immunosuppressive and antifibrotic therapy, the patient experienced progressive decline in lung function (Supplementary Fig. 1a) and was therefore considered a candidate for lung transplantation (LuTX) as a late-stage therapeutic option. During the Omicron wave, the patient became SARS-CoV-2 positive with initially mild symptoms and was treated with the monoclonal SARS-CoV-2-specific antibody sotrovimab (Fig. 1a and Supplementary Table 1 summarizes the indication and mode of action of the drugs described in the manuscript). Over five weeks he experienced progressive respiratory and clinical deterioration requiring hospitalization and subsequent transfer to the Intensive Care Unit (ICU) of the University Medical Center Freiburg, Germany (Fig. 1a). The therapeutic approach included the additional administration of sotrovimab, tocilizumab, and dexamethasone according to the in-house standard of care (Fig. 1a). Upon admission, the severely hypoxemic ($SaO_2/FiO_2 = 70–100$) patient fulfilled the criteria for severe ARDS[1] with a radiological picture of exacerbated ILD (Fig. 1b).

We considered LuTX as the only therapeutic option, anticipating a high mortality associated with SARS-CoV-2-ARDS on underlying SSc-ILD; all involved physicians agreed to LuTX listing because of (i) lacking cure (ii) anticipated high mortality risk and (iii) a worsened LuTX outcome after prolonged ECMO therapy[16,17]. Upper respiratory tract specimens revealed a decreased viral load without cultivable virus (Fig. 1a, c) and the patient received a double LuTX (recipient: HLA-A*03, B*08, B*40, C*03, C*07, DRB1*03, DRB1*04, DQB1*02, DQB1*03, DPB1*01, DPB1*02; donor: HLA-A*01, A*25, B*27, B*37, C*02, C*06, DRB1*01, DRB1*14, DQB1*05, DPB1*04) with perioperative ECMO support. The immunosuppressive regimen included high-dose prednisolone, basiliximab, mycophenolate mofetil (MMF), and tacrolimus according to the local standard (Fig. 1d and Supplementary Table 1)[18], with perioperative anti-infectious therapy including piperacillin/combactam, acyclovir, and sotrovimab. ECMO and ventilator support were stopped on days 1 and 6 post-transplantation, respectively.

### SARS-CoV-2 management after lung transplantation

Despite the non-cultivable viral status before transplantation, SARS-CoV-2 was able to infect the allograft organ and was detectable at day 1 in the URT as well as in the first samples of the LRT taken at day 4 (Fig. 1a). All the URT and LRT samples were tested for other viral and bacterial infections using multiplex PCRs. Bronchoalveolar lavage (BAL) samples were further cultivated for bacterial and fungal growth. No pathogens other than SARS-CoV-2 were detected during this extensive screening. Treatment with remdesivir and the anti-SARS-CoV-2 monoclonal antibodies (mAbs), tixagevimab/cilgavimab, only temporally reduced the viral load, still allowing viral persistence for more than 150 days after LuTX (Fig. 1a). High-resolution computed tomography (HRCT) at day 32 revealed mild ground-glass opacities (GGO), which are generally signs of infection, interstitial lung disease, and/or pulmonary edema[19,20] (Supplementary Fig. 1b). Owing to the clinical well-being of the patient, we interpreted the detection of SARS-CoV-2 as colonization rather than infection. Approximately 85 days after lung transplantation, the patient developed malaise, fever, and increased inflammatory parameters (Supplementary Fig. 2a, b) attributable to a now symptomatic SARS-CoV-2 infection rather than colonization with increased GGO and consolidations on HRCT (Supplementary Fig. 1b, d100). No additional bacterial, viral or fungal infection were detected at this time. In line with these findings, BAL revealed CD8+ T cell-dominated alveolitis (Supplementary Fig. 2c, d). MMF was paused to allow a better lymphocyte response, and remdesivir treatment was repeated for 10 days, resulting in a decreased viral load in the URT/LRT (Fig. 1a, d) without complete viral clearance. Sufficient wound healing and good anastomoses allowed an everolimus-based immunosuppressive strategy, achieving subsequent lower tacrolimus drug levels[21], hypothesizing that an enhanced immune response is a prerequisite for viral clearance[22]. To avoid worrisome drug interactions between nirmatrelvir/ritonavir and immunosuppressive drugs, in addition to the lack of effect of remdesivir, we decided to use molnupiravir as an antiviral alternative. With this therapy and re-administration of tixagevimab/cilgavimab, the patient cleared SARS-CoV-2 in the URT with persistence in the LRT (Fig. 1a). Adjusted treatment, including repeated molnupiravir administration, led to complete SARS-CoV-2 clearance. During follow-ups for more than 18 months after LuTX, the patient had stable lung function, presented clinically well, and remains SARS-CoV-2 free.

### Compartment-specific SARS-CoV-2 evolution for over 140 days

Given the evidence that during prolonged viral replication in immunosuppressed individuals, SARS-CoV-2 acquires mutations and can adapt to antiviral therapies, we analyzed intra-host viral evolution to identify mutations potentially enabling prolonged SARS-CoV-2 infection due to treatment failure or immune escape[11]. Phylogenetic analysis showed that the Omicron BA.2 variant sequences identified after LuTX formed a monophyletic group in the context of all regional Omicron BA.2 variant sequences during the time of infection. Sequences were closely related to the day −9 sequence, confirming the persistence of replicable virus during the time of LuTX despite non-cultivable virus ex vivo (Fig. 2a). During the first 100 days, the virus constantly acquired one mutation every 10 days, mimicking the evolutionary rate of the virus within the general population (Fig. 2b). The observed divergent evolution of viruses isolated from URT and LRT (Fig. 2a and Supplementary Fig. 3a) indicates a milieu-specific selection pressure. The day 141 variant acquired 39 novel mutations within 30 days, and thus had an increased mutation rate of 13 mutations per 10 days, which was temporally associated with molnupiravir treatment (Fig. 2b). Mechanistically, molnupiravir causes lethal mutagenesis of the viral genome by introducing transitions[23]. Analysis of the transition-to-transversion ratio (ts/tv) of the novel day 141 mutations showed a high ts/tv of 22.5, far exceeding expected values and therefore causally linked with the molnupiravir therapy[24]. In addition, half of the mutations were synonymous, indicating an undirected (drug-induced) evolution rather than selection pressure (Fig. 2c and Supplementary Fig. 3b).

As the viral infection was not cleared by the administered SARS-CoV-2-specific neutralizing mAbs, we performed an in-depth analysis of the mutational profile of the viral genome. We focused on the SARS-CoV-2 spike protein to analyze the potential impact of the mutations on antibodies (sotrovimab and cilgavimab/tixagevimab) and ACE2 binding. Interestingly, we identified three non-Omicron BA.2 variant-defining mutations (K356T, L368I, and T385I) that were present on day −9 (Supplementary Table 2). As the patient was already SARS-CoV-2 positive for over 40 days prior to lung

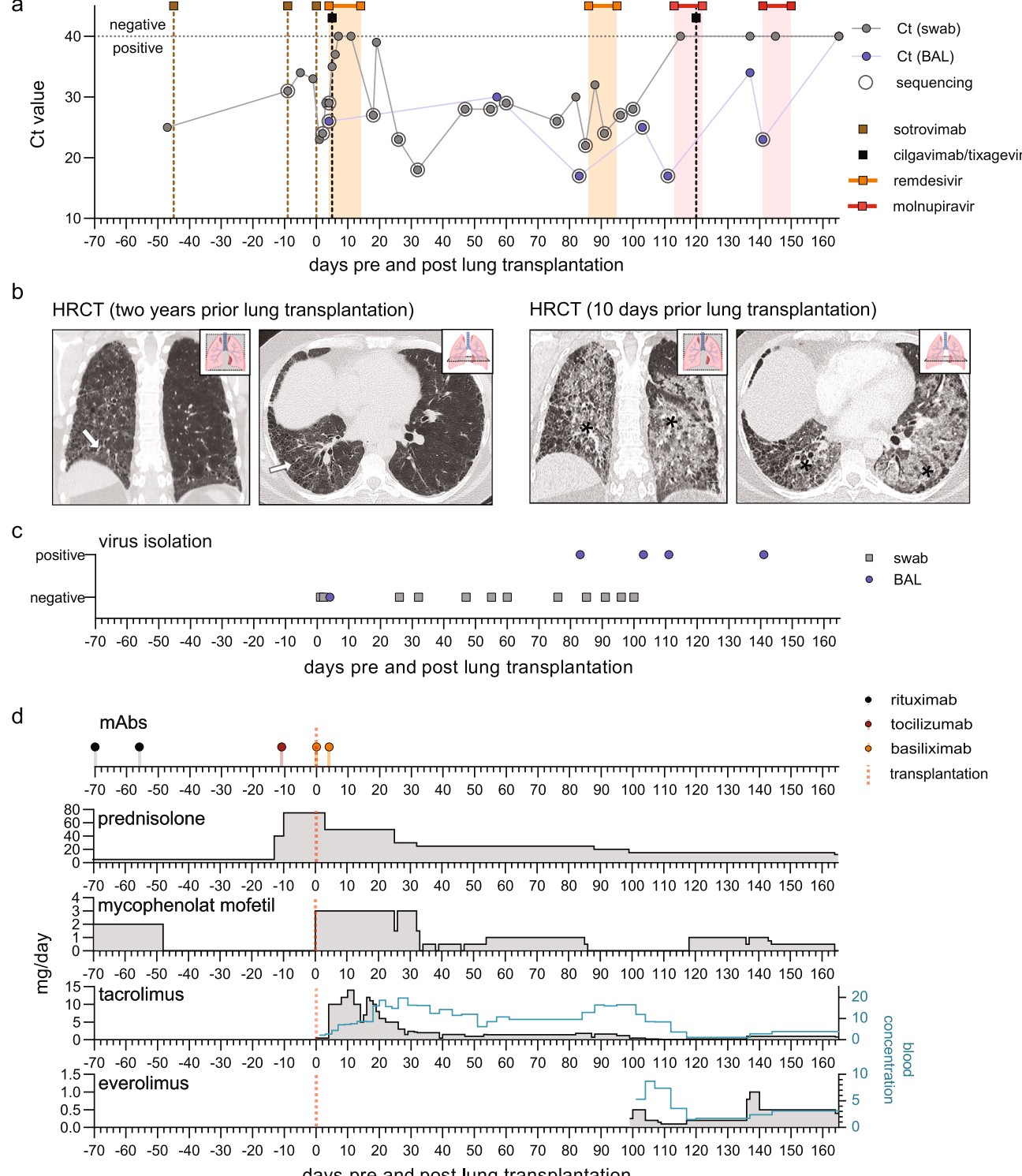

**Fig. 1 | Temporal overview of clinical parameters.** Day 0 indicates the time of lung transplantation. **a** Diagnostic SARS-CoV-2 qPCR cycle threshold (Ct) values of oropharyngeal swabs (gray) and bronchoalveolar lavage (BAL; blue). The horizontal dotted lines indicate the cut-off value (Ct ≥ 40) between positive and negative results. Treatment regimens with SARS-CoV-2-directed antivirals (200 mg once followed by 100 mg/day remdesivir or 2 × 800 mg/day molnupiravir) or single doses of SARS-CoV-2-specific antibodies (500 mg sotrovimab or 150 mg tixagevimab/cilgavimab) are indicated as highlighted regions and vertical lines, respectively. Circles indicate the sequenced patient samples. **b** HRCT two years and 10 days prior to lung transplantation. Arrows indicate subtle reticulations corresponding to underlying interstitial lung disease. Asterisks denote ground glass opacities and consolidations as a surrogate of acute inflammation superposed on underlying interstitial lung disease. Lung icon adapted from BioRender. Fuchs, J. (2025) https://BioRender.com/hv8kbn1. **c** Virus isolation positive cell culture with successful full-genome sequencing of SARS-CoV-2 after the initial passage. **d** Immunosuppressive regimens with prednisolone, mycophenolat mofetil, tacrolimus and everolimus and mAbs (1 g/dose rituximab, 800 mg/dose tocilizumab and 20 mg/dose basiliximab). For the tacrolimus and everolimus, the measured blood concentration (ng/L; light blue line) is shown on the right y-axis.

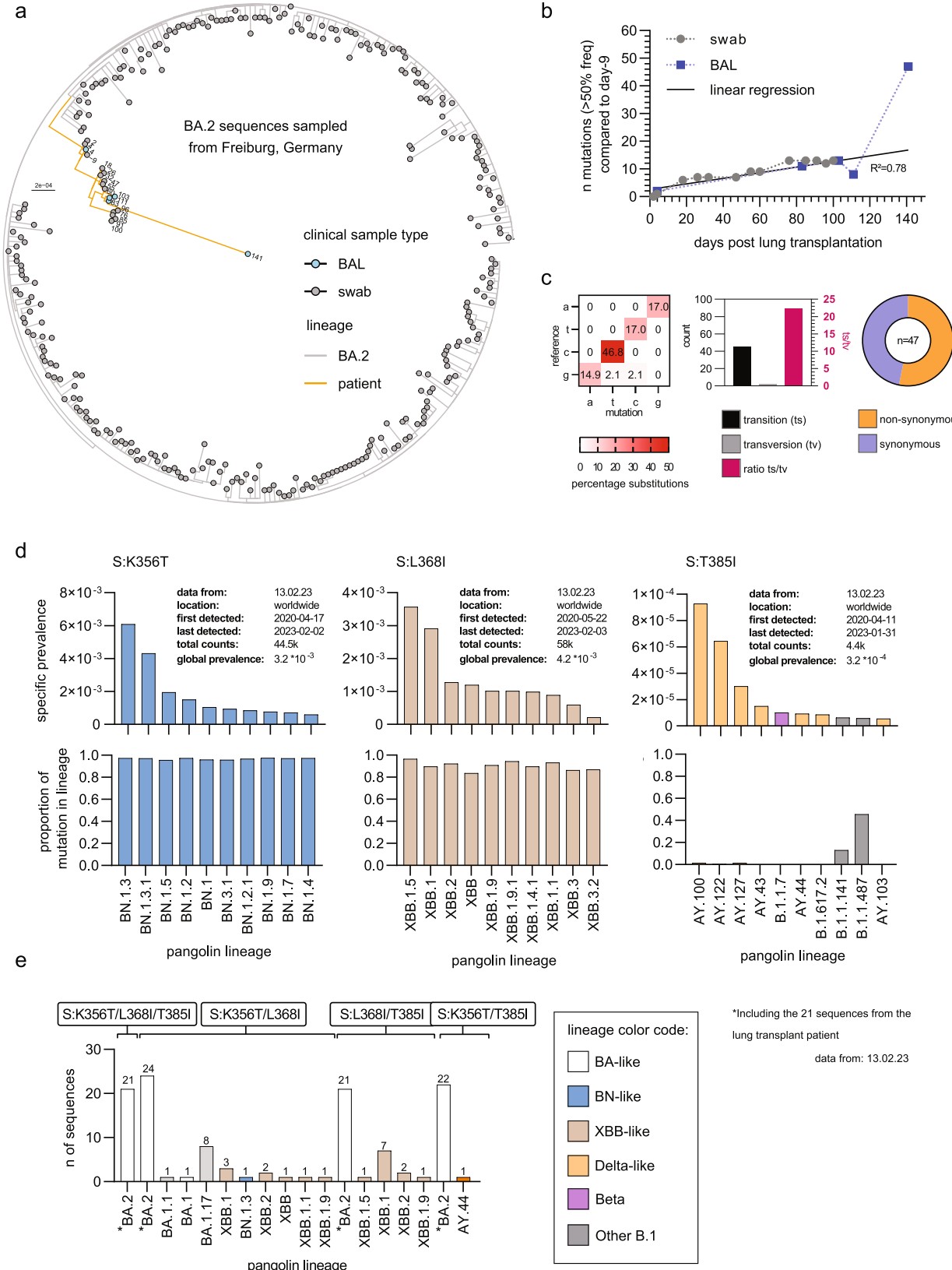

transplantation, we explored the rarity of these mutations in a dataset of 6.9 million SARS-CoV-2 genomes to estimate if these mutations could have developed in-host during the timeframe prior to transplantation. Remarkably, all three mutations were very rare and were mostly present in lineages that emerged later during the pandemic (Fig. 2d). Dual combinations of these mutations were even

rarer, and all three together were found only in the sequences of our patient (Fig. 2e). While it remains impossible to dissect if these mutations were present in the initial virus the patient was infected with, their combinatorial rarity could point to in-host development prior to day −9. We identified a single in-host acquired mutation within the receptor-binding domain (RBD) between day −9 and day

**Fig. 2 | Intra-host evolution of SARS-CoV-2. a** Phylogenetic analyses of the patient SARS-CoV-2 sequences in the context of Omicron BA.2 variant sequences from Freiburg, Germany (Supplementary Table 2). The maximum-likelihood phylogenetic tree was constructed with IQ-Tree (1000 bootstrap replicates, ModelFinder: GTR + F + R2) and rooted to the Wuhan-Hu-1 reference sequence (NC_045512). The tree was visualized with the R ggtree package. Bar indicates substitutions per site. **b** Count of viral mutations with a variant frequency > 50% compared to mutations already present at day -9 plotted against time. Linear regression was performed, excluding day 141. **c** Percentage of substitutions (left), substitution type (middle)

and amino acid effect (right) of novel viral mutations with a variant frequency > 50% of the day 141 sequencing result compared to day -9. **d, e** The outbreak info R package was used to access and analyze 6.9 million SARS-CoV-2 genomes between 2021-12-01 and 2022-10-24 harboring the mutations K356T, L368I and T385I. **d** Specific prevalence (number of lineage sequences harboring the respective mutation/total number of analyzed sequences) of the top 10 lineages where the single mutations have been detected (upper plot) and the proportion (lower plot) of these mutations within the respective lineage. **e** Total sequence count of all sequences that have combinations of the indicated mutations within the dataset.

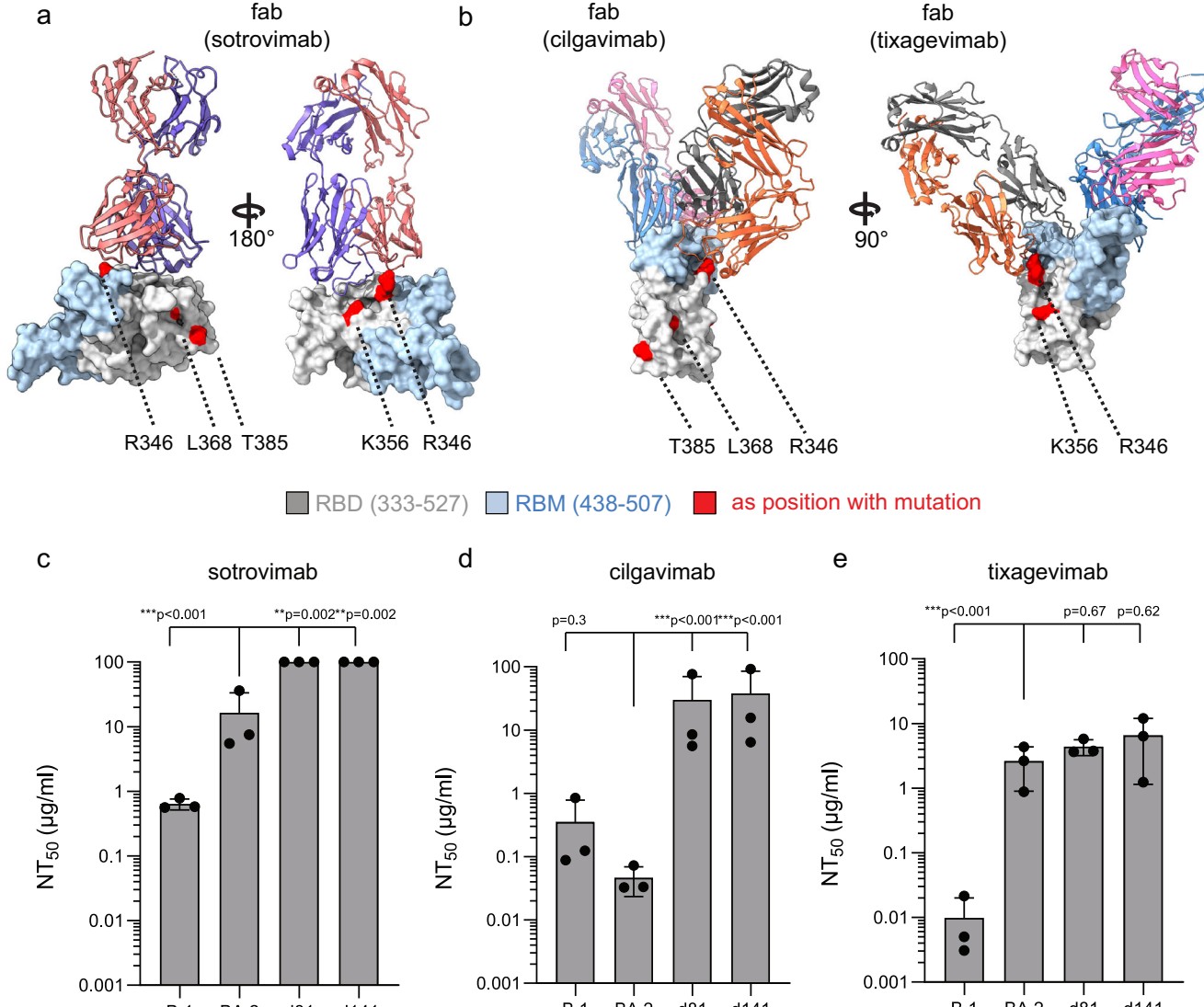

**Fig. 3 | Sensitivity of patient isolates to SARS-CoV-2 specific monoclonal antibodies.** 3D representations of (**a**) the Omicron BA.1 variant receptor-binding domain (RBD) in complex with the fab fragment of sotrovimab (pdb accession 7X1M.) or (**b**) the S RBD in complex with the fabs of cilgavimab and tixagevimab (pdb accession: 7L7E). The mutational sites at position R346, K356, L368 and T385 are highlighted in red. **c**–**e** Neutralizing capacity of the therapeutic antibodies (**c**) sotrovimab, (**d**) cilgavimab and (**e**) tixagevimab. Serial 10-fold dilutions of the

monoclonal antibodies were incubated with 100 pfu of the prototypic wildtype Omicron B.1 variant, a prototypic Omicron BA.2 variant or the two patient Omicron BA.2 isolates (d81 and d141) and analyzed by plaque assay. Neutralization titers 50 ($NT_{50}$) values were calculated from individual curve fits of each serial dilution (3 biologically independent experiments). Shown are the mean and standard deviation. Statistics were performed on log-transformed values with a one-way ANOVA (Tukey's multiple comparison test, **$p \leq 0.01$, ***$p \leq 0.001$).

141. R346T appeared on day 73 post-transplantation and was subsequently detected in URT swabs and BAL samples.

This mutation also became globally prevalent in subsequent Omicron strains, implying convergent evolution into circulating strains[25]. Structural analyses indicated that all four spike (S) mutations lie outside the interface to ACE2, likely not affecting direct receptor binding (Supplementary Fig. 4a, b). However, structural analysis of the

K356T and R346T mutations predicted potential interference with sotrovimab binding, while R346T could interfere with cilgavimab binding (Fig. 3a, b). In line with these in-silico predictions, neutralization assays support strong viral escape of day 81 and day 141 isolates from neutralization by sotrovimab and cilgavimab compared to a prototypic Omicron BA.2 variant isolate (Fig. 3c, d). All Omicron BA.2 variant isolates were already insensitive to tixagevimab (Fig. 3e), as

previously reported[26]. SARS-CoV-2-specific monoclonal antibody treatment likely directed part of viral evolution, causing the development of therapy-resistant viruses. In contrast, we did not observe viral adaptations to remdesivir, suggesting that remdesivir treatment failure was not caused by viral adaptation to the drug. Notably, viruses isolated at days 81 and 141 replicated to comparable titers in Calu3 and VeroE6 cells and only showed a minor replication fitness loss at 72 h post-infection compared to the prototypic Omicron BA.2 variant isolate (Supplementary Fig. 5). This observation implies that in the present case, a single 10-day treatment cycle with molnupiravir was not sufficient to completely clear the viral infection, as a severely mutated but replication-competent virus persisted in the transplanted lung (Fig. 1a).

In summary, antiviral SARS-CoV-2-specific treatment with mAbs affected viral evolution and led to initial treatment failure. Subsequent molnupiravir treatment did not abolish active viral replication during the first treatment cycle.

### Immune response towards SARS-CoV-2

The delayed viral clearance in the donor lung compared to the recipient URT prompted us to investigate the SARS-CoV-2-specific effector arm of the adaptive immune response in the context of a complete HLA-mismatch. SARS-CoV-2-specific anti-N IgG or anti-N/S IgM levels in serum samples remained below the detection limit during the time of infection (Fig. 4a). This observation indicates the absence of a virus-specific humoral response, likely attributed to the preceding B cell-depleting therapy with rituximab. In contrast, SARS-CoV-2-specific CD4+ and CD8+ T cells were present in this patient (Supplementary Fig. 6a, b). Circulating SARS-CoV-2-specific CD4+ T cells were directed towards recipient-specific epitopes and SARS-CoV-2-derived epitopes that were restricted by multiple HLA class II alleles, so-called promiscuous epitopes (Supplementary Fig. 6a). Consequently, these epitopes can be recognized by CD4+ T cells from both donor and recipient. However, we only recovered CD4+ T cells from the patient (recipient, HLA-A*03+) and not from the donor in blood or BAL (Supplementary Fig. 6c). Circulating SARS-CoV-2-specific CD8+ T cells targeted a broad epitope repertoire comparable to that of healthy SARS-CoV-2 convalescent controls, showed similar frequencies, and exhibited a robust functional profile including cytokine production (IFN-γ and TNF) and degranulation (surrogate marker: CD107a) upon SARS-CoV-2-specific peptide stimulation (Fig. 4b–d and Supplementary Fig. 7). In the blood, only recipient SARS-CoV-2-specific CD8+ T cells (restricted by HLA-A*03, HLA-B*40) were detectable in the absence of donor SARS-CoV-2-specific CD8+ T cells (restricted by HLA-A*01, HLA-B*27) (Fig. 4c and Supplementary Fig. 6b). Thus, we did not observe T cell chimerism in the circulation, which is in line with previous reports[27,28]. In contrast, the BAL samples, reflecting the alveolar compartment of the HLA-mismatched lung transplant, exhibited chimerism, containing both donor- and recipient-derived SARS-CoV-2-specific CD8+ T cells (Supplementary Fig. 7). However, the frequency of donor SARS-CoV-2-specific CD8+ T cells was lower than that of the recipient cells (Fig. 4e). At follow-up, these donor T cells were not detectable (Fig. 4e), suggesting that they were derived from a previous donor infection rather than from de novo induction by SARS-CoV-2 infection upon LuTX. In addition, donor T cells expressed higher levels of αE integrin CD103 and showed increased co-expression of CD69 and CD103 compared to recipients' T cells until they declined below the detection limit (Fig. 4f and Supplementary Fig. 8), indicating tissue-resident memory characteristics. This observation also supports the hypothesis that SARS-CoV-2-specific memory T cells are transferred from the donor to the recipient within the transplanted organ and contribute to the local immunity. Based on the finding that the SARS-CoV-2-specific CD8+ T cell response was present in the blood and alveolar compartments without viral clearance, we wondered whether viral evolution resulted in viral escape from T cell recognition.

Sequence analyses of the HLA-restricted viral T cell epitopes of donor and recipient did not reveal adaptive mutations, rendering viral escape from T cell recognition unlikely (Supplementary Fig. 9). Intrigued by the observation of diverged viral clearance in the recipient's URT and the donor's LRT, we hypothesized that HLA-mismatch may impair the recipient's SARS-CoV-2-specific CD8+ T cells to control the infection in the donor lung. There was no overlap of the HLA class I and II alleles between the recipient and donor, and we did not observe cross-reactivity of the circulating recipient SARS-CoV-2-specific CD8+ T cell populations (Supplementary Fig. 6b). Thus, SARS-CoV-2-infected donor lung tissue was invisible to the recipient's CD8+ T cell response, which may also explain the faster viral clearance in the URT. Taken together, failure of the effector arm of the immune system dictated prolonged SARS-CoV-2 infection, that is (i) drug-induced B cell depletion impaired antibody formation and (ii) HLA class I disparity caused CD8+ T cell blindness in SARS-CoV-2-infected HLA-mismatched donor lung tissues.

## Discussion

During the COVID-19 pandemic, antiviral treatment has been constantly changing due to additional findings and the emergence of variants of concern[29–31]. Treatment of persistent SARS-CoV-2 infections in immunocompromised patients is challenging due to a weakened immune response and the risk of developing therapy-resistant mutations during prolonged infection[11,32]. In particular, B-cell-depleted patients were reported to suffer from SARS-CoV-2 persistence[14,33,34].

We describe successful lung transplantation in a patient without complete SARS-CoV-2 clearance in URT swaps, providing insights into clinical management, viral evolution, and especially enlarging the understanding of immune response in an HLA-disparate organ.

First, this case demonstrates that LuTX was successful and saved the patient's life despite detectable SARS-CoV-2 in the URT and subsequent transmission to the transplant. Current recommendations advise against lung transplantation for patients without cleared viral infections[35] based on a case report in previously healthy individuals. In this case, the advanced SSc-ILD and the immunosuppressive regimen made recovery from SARS-CoV-2-associated ARDS unlikely, and the time to viral clearance unpredictable. Mechanical ventilation and ECMO therapy may complicate or even hamper lung transplantation due to therapy-associated complications[16,17,36]. Therefore, SARS-CoV-2-positive ARDS patients with an underlying lung disease might benefit from earlier LuTX, even risking SARS-CoV-2 transmission to the transplant.

Second, viral evolution may precede standard treatment protocols that include the use of neutralizing antibodies and remdesivir. Viral genome sequencing and structural analyses identified mutations in the viral S protein, allowing viral escape to sotrovimab and cilgavimab, as confirmed by in vitro neutralization assays. We speculate that the therapeutic use of sotrovimab and cilgavimab/tixagevimab in the absence of a SARS-CoV-2-specific immune response facilitated the development of therapy-resistant variants[11]. After lung transplantation, the virus showed a constant mutation rate even under remdesivir therapy, which can increase viral intra-host diversity[37]. We decided against both the use of nirmatrelvir/ritonavir owing to their pharmacological interactions with immunosuppressive drugs[38] and to further remdesivir cycles. Two consecutive molnupiravir cycles and a modified immunosuppressive regime[22] resulted in viral clearance in URT and later in LRT. Mechanistically, molnupiravir causes lethal mutagenesis of the viral genome[24]. However, the first molnupiravir course did not lead to viral clearance, and we isolated a hypermutated but not attenuated virus. While the effects of molnupiravir have been extensively described, recent data showed that there is a global signature of molnupiravir-induced mutations, implying that such hypermutated, but viable variants have been transmitted to the general population[39]. A second study showed hypermutation and

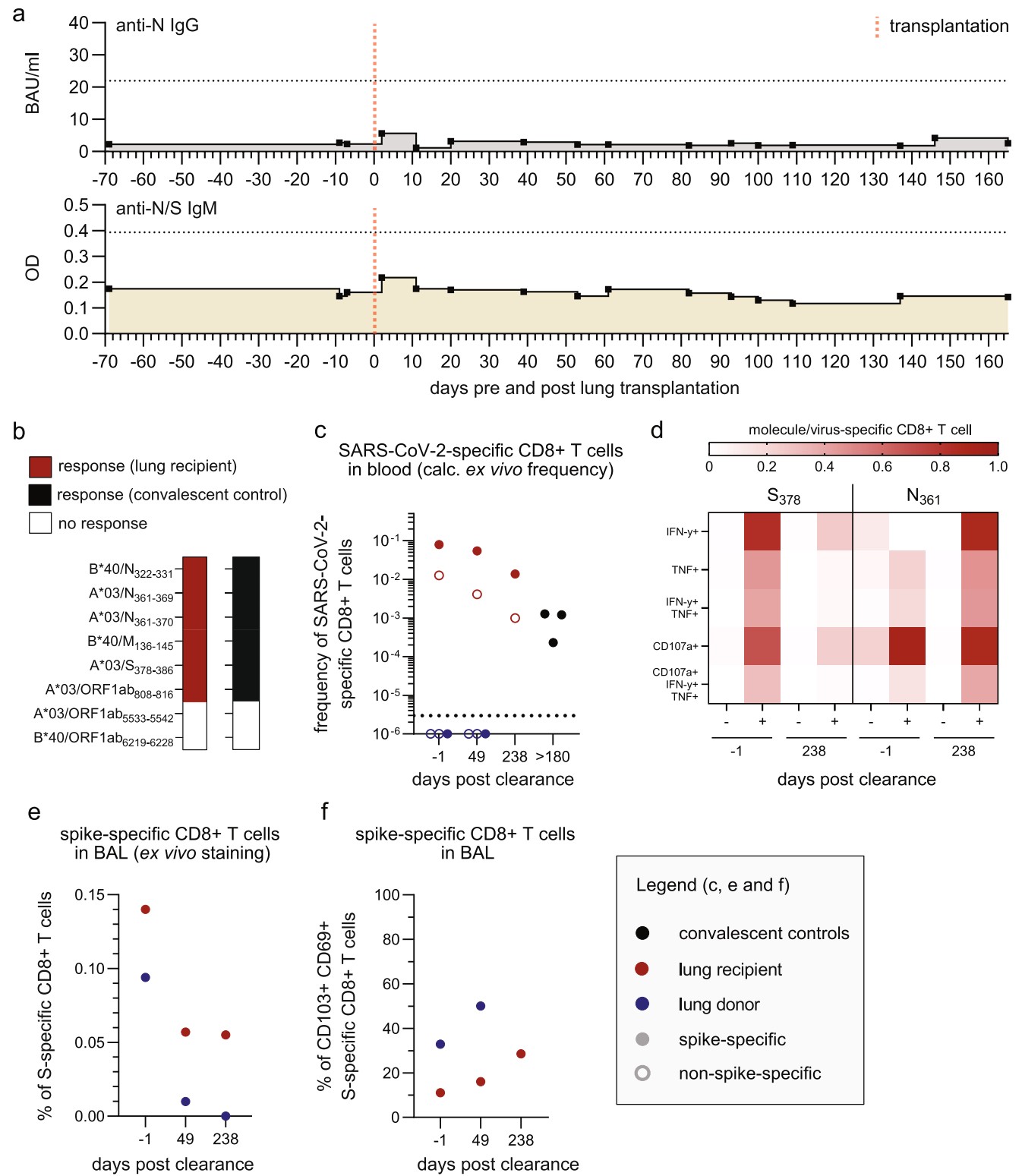

non-clearance in molnupiravir-treated, SARS-CoV-2-infected immunosuppressed patients[40]. However, none of these studies have shown that these viruses are replication-competent. Our case provides clear evidence of molnupiravir-induced hypermutation without viral clearance or compromising viral fitness. We show that these viruses can be isolated and are replication competent in cell culture. This supports the notion that molnupiravir-mutated SARS-CoV-2 variants could, in theory, be transmitted to the general population. Another important aspect of non-synchronized viral clearance is that standard SARS-CoV-

2 diagnostics rely on upper respiratory specimens and would have missed the ongoing infection in the donor graft.

The third finding of this case study is that the paradigm of HLA-restriction for effective CD8+ T cell immunity is relevant in the context of lung transplantation, which is primarily performed in an HLA-mismatched setting. During SARS-CoV-2 infection, T cells mount their antiviral activity at the mucosal and alveolar sites. In the case of an infected transplanted lung with a disparate HLA type, recipient T cells do not recognize SARS-CoV-2-infected cells within the transplanted

**Fig. 4 | SARS-CoV-2 directed adaptive immune response. a** Serological analysis of SARS-CoV-2 N-specific IgG (upper) and SARS-CoV-2 S- and N-specific IgM (lower) by ELISA. Horizontal dotted lines mark the detection limits. **b** Circulating CD8+ T cell responses against epitopes described to be restricted by the HLA-A/B types of the lung transplant recipient and healthy SARS-CoV-2 convalescent control. **c** SARS-CoV-2-specific CD8+ T cells detected in blood samples after pMHCI tetramer-based enrichment. CD8+ T cells targeting 2 to 5 distinct SARS-CoV-2 epitopes were analyzed in blood samples from the lung transplant recipient at three time points. Calculated ex vivo frequencies of A*03/$S_{378}$- (red), A*03/$N_{361}$- (red), A*01/$S_{865}$- (blue), A*01/$ORF3a_{207}$- (blue) and A*01/ $ORF1ab_{4163}$- (blue) specific CD8+ T cells are depicted from samples of lung transplant recipient 1 day before and 49 and 238 days after complete SARS-CoV-2 clearance. For comparison, calculated ex vivo frequencies of A*03/$S_{378}$- (black) specific CD8+ T cells are shown from healthy SARS-CoV-2 convalescent controls ($n = 3$) at >180 days post-infection. Horizontal line marks the detection limit. **d** Production of IFN-γ, TNF and CD107a per SARS-CoV-2-specific CD8+ T cell after 14 days of in vitro expansion and peptide stimulation. IFN-γ, TNF and CD107a production is depicted for unstimulated (-) and peptide-stimulated (+) samples of lung transplant recipients. Samples were collected 1 day before and 238 days after complete viral clearance. **e** Recipient (A*03/$S_{378}$ (red))- and donor (A*01/$S_{865}$ (blue))-derived, spike-specific CD8+ T cells detected in BAL samples of the lung transplant recipient ($n = 1$) 1 day before and 49 and 238 days after complete SARS-CoV-2 clearance using ex vivo pMHCI tetramer staining. **f** CD103 and CD69 expression on recipient (A*03/$S_{378}$ (red)) - and donor (A*01/$S_{865}$ (blue))-derived, spike-specific CD8+ T cells detected in BAL samples of the lung transplant recipient ($n = 1$) at the indicated time points after complete SARS-CoV-2 clearance.

lungs. In contrast, donor T cells, still able to recognize SARS-CoV-2-infected cells, persist in the transplanted lung for months[28,41] and acquire tissue-resident memory phenotypes. These ex vivo observations may explain the non-synchronized viral clearance in the (recipient's) URT and (donor's) LRT, and may be the reason for ineffective viral clearance. Blood sampling is an easy and feasible way to collect circulating immune cells, assess antibody levels and measure cytokine levels; however, it does not necessarily reflect the immune response at the site of infection. In the early phase after lung transplantation, donor T cells are still present to encounter SARS-CoV-2, although low frequencies might be insufficient for effective viral clearance. In line with previous studies[28,41], donor T cells declined to undetectable levels eight months after lung transplantation. This case suggests that during lung transplantation, the transferred immune cells can have a pivotal role, and the host's immune system might be inherently blind in the donor graft. However, our observations were restricted to a single patient and a limited sample material. Therefore, further analyses with more patients, including different HLA-mismatch patterns (based on eplet and supertype classifications) and non-polymorphic HLA-E[42,43], may elucidate the protective benefit of the graft's transplanted immune system. In addition, these investigations may also reveal whether prior infections or vaccinations in the donor confer protection from infection in the recipient.

Even though we have focused in this case report on the role of HLA mismatch in a lung transplant recipient with prolonged SARS-CoV-2 infection, persistence of SARS-CoV-2 has also been shown in other settings. Acquired (e.g., due to B cell-depleting therapies) or innate immunodeficiency contribute to prolonged viral persistence, and beyond impaired T cells, also other immune cells (e.g., B cells, natural killer cells) play a pivotal role in viral clearance[14,33,44–48].

It will be interesting to elucidate how the donor's immune system, primed by previous infection and vaccination, influences post-transplant infections. Strategies to conserve or replenish the immune response compatible with the donor's tissue and, if possible, considering transplantation with at least one overlapping HLA class I allele between donor and recipient may represent pivotal aims to reduce the number and severity of infections after lung transplantation.

## Methods
### Clinical data and biosamples
Clinical data were extracted from medical records. Post-transplant care followed the in-house standard of care[18,49] with adaptation of interventions and therapies to prolonged SARS-CoV-2 infection as depicted below. Biosamples were obtained for routine clinical care. Healthy donor samples were included in Fig. 4b+c as controls ($n = 2$ males (37 and 43 years); $n = 3$ females (24, 25 and 27 years)). Written informed consent was obtained from the patient and healthy donors. The study was conducted in accordance to federal guidelines, local ethics committee regulations (Albert-Ludwigs-Universität, Freiburg, Germany; vote: 322/20, 10/03, 21–1135 and 383/19) and the Declaration of Helsinki (1975).

### Bronchoscopy and bronchoalveolar lavage (BAL)
Bronchoscopy and BAL were performed as surveillance bronchoscopies as usual post-transplant care or as clinically indicated. BAL samples and lymphocyte analyses were performed as described below[50].

### Analysis of the lymphocyte counts
Post transplantation bronchoscopies were performed at the indicated time points (Fig. 1a) as part of the usual post-transplant care and as medically indicated[18]. Brochoalveolar lavage (BAL) was performed in the middle lobe segment or the lingual with 200–300 mL pre-warmed saline in 20 mL aliquots in a bronchoscopic wedge position[50]. Saline solution was recovered by gentle suction and pooled in a polypropylene tube. Lavage fluid was kept on ice and processed immediately by filtering through two layers of cotton gauze. The cells were centrifuged at 500 g and then washed with phosphate-buffered saline (PBS) at 4 °C. Cell count and cell viability were assessed after staining with trypan blue, using a Thoma Chamber (Roth, Germany). For cell differentiation, cells were resuspended in PBS containing 1 % BSA, and cell smears were prepared. Air-dried smears were stained by HEMA-COLORTM (E. Merck, Darmstadt, FRG) and a minimum of 300 cells were counted throughout the whole slide. To determine lymphocyte subpopulations, BAL cells were stained with the antibodies listed in Supplementary Table 3. Stained cells were measured using a CytoFLEX cytometer (Beckman Colter, Krefeld, Germany) and percentages were evaluated using CytExpert software v.2.4.0.28 (Beckman Colter, Krefeld, Germany).

### Virus detection by qPCR
SARS-CoV-2 RNA testing of oropharyngeal swabs was performed using Alinity m SARS-CoV-2 assay (09N78-095, Abbott, Illinois, USA). RNA samples were extracted using the QIAamp MinElute Virus Spin kit (57704, Qiagen, Hilden, Germany). The test was performed and interpreted according to the manufacturer's instructions, and semi-quantitative results reported in cycle threshold (Ct) values.

### Serological testing
SARS-CoV-2-specific anti-nucleoprotein (N) IgG (7304, Mikrogen Diagnostik GmbH, Neuried, Germany) and anti-N/anti-S IgM (ESR400M, Serion, Germany) ELISAs were performed according to the manufacturer's protocol. Results were evaluated semi-quantitatively as arbitrary units (AU) compared to the manufacturer's calibrators or shown as raw values.

### Cell culture, virus isolation and growth kinetics
Virus isolation and cell culture experiments with SARS-CoV-2 were performed under Biosafety Level 3 (BSL3) protocols at the Institute of Virology, Freiburg. Adherent African green monkey kidney VeroE6 cells (ATCC CRL-1586) and human lung Calu-3 cells (ATCC HTB-55), kindly provided by Markus Hoffmann (Göttingen), were cultured in 1 × Dulbecco's modified Eagle medium (DMEM) containing 5% or 10% fetal calf

serum (FCS), respectively. All cell lines were routinely tested for mycoplasma. To isolate SARS-CoV-2 from patient material, filtered throat swabs were inoculated on $2 \times 10^6$ Calu-3 cells in 4 mL DMEM with 2% FCS and incubated at 37 °C and 5% $CO_2$ for 4–6 days until the cytopathic effect was visible. The culture supernatant was cleared and stored at − 80 °C. Virus titers were determined by plaque assay on VeroE6 cells. Mutations in the viral genomes of the initial isolation and all derived virus stocks were confirmed by next-generation sequencing.

For viral growth kinetics $1 \times 10^6$ VeroE6 or Calu-3 cells were infected with a multiplicity of infection of 0.001 for 1.5 h. Cells were washed three times with PBS and overlaid with 2 mL DMEM with 2% FCS. The supernatants were collected at 8 h, 24 h, 48 h and 72 h post infection. Viral titers were determined by plaque assay on VeroE6 cells. As a control, a prototypic BA.2 isolate was used (EPI_ISL_9324096).

### Evaluation of the neutralizing capacity of therapeutic monoclonal antibodies
Neutralizing antibody titers were determined by a plaque reduction assay. Therefore, serial monoclonal antibodies dilutions were incubated with 100 plaque-forming units (pfu) of the SARS-CoV-2 isolates for 1 h. The mixture was dispersed on VeroE6 cells in a 12-well format, and cells were overlaid with 0.6% oxoid-agar for 72 h at 37 °C. Fixed cells were stained with 0.1% Crystal violet. The number of plaques was compared with an untreated control without serum or antibodies. To evaluate the neutralizing capacity and determine the neutralizing titer 50 ($NT_{50}$), a non-linear fit least squares regression (constraints: bottom constant equal to 0 and upper constant equal to 100) was performed. Besides the patient isolates and the prototypic BA.2 isolate, the Muc-IMB-1 isolate (lineage B.1) was used as a control (EPI_ISL_406862 Germany/BavPat1/2020)[51,52], kindly provided by Roman Woelfel, Bundeswehr Institute of Microbiology.

### Whole-genome sequencing
For SARS-CoV-2 sequencing, the NEBNext ARTIC SARS-CoV-2 FS Library Prep Kit (E7658L, NEB, Frankfurt am Main, Germany) was used. Briefly, cDNA was generated from the RNA of oropharyngeal swabs (57704, QIAamp MinElute Virus Spin kit, Qiagen) or from cell culture supernatants (R1035, Quick-RNA Viral Kit, Zymo Research). The viral genome was then amplified by PCR with primers tiling the entire viral genome. Subsequently, indexed paired-end libraries for Illumina sequencing were prepared. Normalized and pooled sequencing libraries were denatured with 0.2 N NaOH and sequenced on an Illumina MiSeq instrument using the 300-cycle MiSeq Reagent Kit v2 (MS-102-2002, Illumina).

The de-multiplexed raw reads were subjected to a custom Galaxy pipeline, which is based on SARS-CoV-2 analysis pipelines on usegalaxy.eu[52] The raw reads were pre-processed with fastp[53] and mapped to the SARS-CoV-2 Wuhan-Hu-1 reference genome (Genbank: NC_045512) using BWA-MEM[54]. Primer sequences were trimmed with ivar trim (https://andersen-lab.github.io/ivar/html/manualpage.html). Variants (SNPs and INDELs) were called with the ultrasensitive variant caller LoFreq[55], demanding a minimum base quality of 30 and a coverage of at least 20-fold. Afterwards, the called variants were filtered based on a minimum variant frequency of 10% and on the support of strand bias. The effects of the mutations were automatically annotated in the vcf files with SnpEff[56]. Finally, consensus sequences were constructed by bcftools (v.1.10)[57]. Regions with low coverage or variant frequencies between 0.3 and 0.7 were masked with Ns. Final consensus sequences have been deposited in the GISAID database (www.gisaid.org) (Supplementary Table 4). Raw data has been deposited in the European Nucleotide Archive (ENA) under the study accession number: PRJEB71389.

### Phylogenetic analysis
All Omicron BA.2 variant sequences generated in Freiburg, Germany between the 01.03.2022 and 10.11.2022 were included in the analysis

(Supplementary Table 4). Then, a maximum-likelihood phylogenetic tree was constructed based on SARS-CoV-2 full genome consensus sequences. Therefore, sequences were aligned with MAFFT (v7.45)[58] and a tree was constructed with IQ-Tree (v2.1.2)[59]. The best-fitting substitution model was automatically determined, and the tree was calculated with 1000 bootstrap replicates. Branch support was approximated using the Shimodaira–Hasegawa [SH]-aLRT method (1000 replicates). The tree was rooted to the reference sequence NC_045512. To visualize the phylogenetic tree, a custom R script was written utilizing the ggtree (v2.2.4)[60], treeio (v1.12.0)[61] and ggplot2 (v3.3.3) packages.

### Mutational analysis
An in-house R script was used to plot the variant frequencies that were detected by LoFreq as a heatmap (pheatmap package v1.0.12). The script is publicly available (github.com/jonas-fuchs/SARS-CoV-2-analyses, v.1.1 https://doi.org/10.5281/zenodo.7692398) and has also been implemented as a Galaxy tool (Variant Frequency Plot on usegalaxy.eu). Percentage of substitutions, substitution type and amino acid effect of novel viral mutations with a variant frequency > 50% compared to Wuhan-Hu-1 were calculated on the basis of the annotated VCF files.

To analyze the global number of sequences that harbor the single or combinatorial spike mutations (S:K356T, S:L368I, S:T385I), GISAID was accessed with the outbreakinfo R package on the 13.02.2023, including all sequences between the 12.01.2021 and 24.10.2022 (~ 6.9 million sequences)[62].

BA.2 consensus mutations were calculated on the basis of 4992 BA.2 sequences from Baden-Wuerttemberg, Germany, between April and December 2022 (GISAID Identifier: EPI_SET_230216es, https://doi.org/10.55876/gis8.230216es). Mutational profiles were determined with covsonar v.1.1.8 (https://github.com/rki-mf1/covsonar) and a lightweight Python script (https://github.com/jonas-fuchs/covsonar_con_mut).

### Visualization of the spike protein structure
The EM structures were accessed from the protein data bank (7XIX, 7XB0, 7X1M, 7L7E) and visualized with UCSF ChimeraX version: 1.1 (2020-09-09).

### Peripheral blood mononuclear cell (PBMC) isolation
PBMCs were isolated from venous blood samples and BAL by lymphocyte separation medium density gradients (Pancoll separation medium, PAN Biotech GmbH) and either stored at − 80 °C until further processing or immediately used for further analyses.

### Peptides and tetramers for SARS-CoV-2-specific T cell analysis
Peptides were produced with an unmodified N-terminus and an amidated C-terminus with standard Fmoc chemistry and exhibited > 70% purity (Genaxxon Bioscience, Ulm, Germany) (Supplementary Table 5). For tetramer generation, first peptides were loaded on biotinylated HLA class I easYmers (HLA-A*01:01 (1001-01), HLA-A*03:01 (1016-01), immunAware, Hørsholm, Denmark). Subsequently, the peptide-loaded HLA class I easYmers were tetramerized by conjugation to allophycocyanin- (APC, BD Biosciences, Heidelberg, Germany) or phycoerythrin- (PE, Agilent, California, US) coupled streptavidin according to the manufacturer's instructions (https://immunaware.com/wp-content/uploads/2024/08/HLA-easYmers%C2%AE-Full-protocol-1.pdf).

### Detection of SARS-CoV-2-specific CD8+ T cells in blood and BAL samples
SARS-CoV-2-specific ($A^*03/S_{378-386}$, $A^*03/N_{361-369}$, $A^*01/S_{865-873}$, $A^*01/ORF3a_{207-215}$, $A^*01/ORF1ab_{4163-4172}$) CD8+ T cells in blood samples were identified by magnetic bead-based enrichment[63,64]. To this end, $8 \times 10^6$ to $10 \times 10^6$ PBMCs were stained with APC-conjugated peptide-loaded HLA class I tetramers for 30 min. Subsequently, PBMCs were washed

with PBS including 0.5 % BSA and 2 mM EDTA, and incubated with magnetic anti-APC microbeads (Miltenyi Biotec, Bergisch Gladbach, Germany) for 20 min at 4 °C. Afterwards, the cells were washed again and then subjected to positive selection using MACS technology (Miltenyi Biotec, Bergisch Gladbach, Germany) according to the manufacturer's instructions (https://static.miltenyibiotec.com/asset/150655405641/document_rcctpg55vh435a1ffh7gfbac14?content-disposition=inline). SARS-CoV-2 spike-specific (A*03/$S_{378-386}$, A*01/$S_{865-873}$) CD8+ T cells in BAL samples were detected by ex vivo staining with fluorophore-coupled peptide-loaded HLA class I tetramers. All samples were analyzed by flow cytometry using the antibodies listed in Supplementary Table 3. After fixation of the cells in 2% paraformaldehyde (PFA, Sigma, Taufkirchen, Germany), analyses were performed on a CytoFLEX (Beckman Colter, Krefeld, Germany) with CytExpert Software v.2.3.0.84. Data were further analyzed using FlowJo v.10.7.1 (Treestar).

### In vitro expansion of SARS-CoV-2-specific T cells and intracellular IFN-γ staining

For expansion and stimulation of SARS-CoV-2-specific T cells, predescribed minimal epitopes or for the HLA types best matching, predicted epitopes were used (Supplementary Table 5). For in vitro expansion of SARS-CoV-2-specific CD8+ T cells, 20 % of the PBMCs were first stimulated with a pool of optimal epitopes (10 µg/mL) for 1 h at 37 °C[65]. Subsequently, the cells were washed, and the remaining, unstimulated PBMCs were added in RPMI medium supplemented with interleukin-2 (IL-2; 20 U/mL, StemCell Technologies). For expansion of SARS-CoV-2-specific CD4+ T cells, $2 \times 10^6$ PBMCs were stimulated with a pool of 3 to 4 peptides (10 µM) and anti-CD28 monoclonal antibody (0.5 µg/mL, Pharmingen Becton Dickinson, Heidelberg, Germany). SARS-CoV-2-specific CD4+ and CD8+ T cells were expanded for 14 days in complete RPMI cell culture medium. On day 14, PBMCs were restimulated with individual peptides, left untreated as a negative control or stimulated with phorbol 12-myristate 13-acetate (PMA) and ionomycin as a positive control in the presence of brefeldin A (GolgiPlug, 0.5 µl/mL; BD Biosciences, Heidelberg, Germany) and IL-2. After 5 h of incubation at 37 °C, surface and intracellular IFN-γ staining was performed. T cell responses were determined by subtracting the signal detected in unstimulated samples from stimulated samples and subsequently applying a cut-off of 0.01 %. For flow cytometry analysis, the antibodies listed in Supplementary Table 3 were used. After fixation of the cells in 2% PFA, analyses were performed on a FACSCanto system (BD Biosciences) with the FACSDiva software v.10.6.2 (BD Biosciences). Data were further analyzed using FlowJo v.10.7.1 (Treestar).

### In vitro expansion of SARS-CoV-2-specific CD8+ T cells and analyses of effector function

A*03/S378-386- and A*03/N361-369-specific CD8+ T cells were expanded by stimulation of $1.5 \times 10^6$ PBMCs with the respective peptide (10 µM) and anti-CD28 monoclonal antibody (0.5 µg/mL, Pharmingen Becton Dickinson, Heidelberg, Germany). SARS-CoV-2-specific T cells were expanded for 14 days in complete RPMI cell culture medium. On day 14, expanded SARS-CoV-2-specific CD8+ T cells were restimulated with the respective peptide, left untreated as a negative control or stimulated with PMA and ionomycin as a positive control[66]. After 1 h of incubation at 37 °C, brefeldin A (GolgiPlug, 0.5 µl/mL) and monensin (GolgiStop, 0.5 µl/mL) (both BD Biosciences, Heidelberg, Germany) were added, and cells were incubated for an additional 4 h at 37 °C. After in total 5 h of incubation, surface and intracellular cytokine staining was performed. In addition to the restimulation, expanded cells were stained with peptide-loaded HLA class I tetramers to assess the functionality of the expanded cells. For flow cytometry analysis, the antibodies listed in Supplementary Table 3 were used. After fixation of the cells in 2 % PFA (Sigma, Taufkirchen, Germany), analyses

were performed on a CytoFLEX (Beckman Colter, Krefeld, Germany) with CytExpert Software v.2.3.0.84 and a FACSCanto system (BD Biosciences, Heidelberg, Germany) with the FACSDiva software v.10.6.2 (BD Biosciences, Heidelberg, Germany). Data were further analyzed using FlowJo v.10.7.1 (Treestar).

### Detection of CD4+ T cells in blood and BAL samples

For the detection and differentiation of donor and recipient CD4+ T cells, PBMCs and BAL samples were stained ex vivo with the antibodies listed in Supplementary Table 3. After fixation of the cells in 2% PFA (Sigma, Taufkirchen, Germany), analyses were performed on LSRFortessa with FACSDiva software v.10.6.2 (BD Biosciences, Heidelberg, Germany). Data were further analyzed using FlowJo v.10.7.1 (Treestar).

### Alignment of patient SARS-CoV-2 sequence to T cell epitope sequences

To analyze whether T cell epitopes are affected by viral mutation, amino acid sequences of CD8+ and CD4+ T cell epitopes, which have been either described or predicted to be restricted by the HLA types of lung donor and recipient, were mapped to SARS-CoV-2 sequences isolated from the lung transplant recipient during the course of infection. These sequence analyses were performed in Geneious Prime 2022.0.2 Clustal Omega 1.2.2 alignment with default settings[67].

### Data analysis

Data analyses, statistics and plotting were performed with GraphPad Prism v8.4.2, R Studio (R version 4.2.1) or Python 3.9.

### Statistics & reproducibility

No statistical method was used to predetermine the sample size. No data were excluded from the analyses. The experiments were not randomized. The Investigators were not blinded to allocation during experiments and outcome assessment.

### Reporting summary

Further information on research design is available in the Nature Portfolio Reporting Summary linked to this article.

## Data availability

SARS-CoV-2 consensus sequences have been deposited in the GISAID database and are publicly available (www.gisaid.org) (Supplementary Table 2). Raw sequencing data has been deposited in the European Nucleotide Archive (ENA) under the study accession number: PRJEB71389. The raw values for charts and graphs are available in the Source Data file whenever possible. All requests for additional raw (especially flow cytometry data) and materials are promptly reviewed by the University of Freiburg Center for Technology Transfer to verify if the request is subject to any intellectual property or confidentiality obligations. Donor-related data not included in the paper were generated as part of clinical examination and may be subject to donor confidentiality. Any data and materials that can be shared will be released via a Material Transfer Agreement. Source data are provided in this paper.

## Code availability

The script to visualize the variant frequencies and phylogenetic trees is publicly available (github.com/jonas-fuchs/SARS-CoV-2-analyses, v1.1, https://zenodo.org/badge/latestdoi/336032336), and the heatmap has been additionally implemented on usegalaxy.eu (Variant Frequency Plot).

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

## Acknowledgements

The authors thank the patient for allowing to publish the case. The care for severely diseased patients (i.e., the transplant setting) requires a multidisciplinary and multiprofessional team, and we are grateful for all the medical staff caring for these patients. We do not want to forget the family of the organ donor, who still suffers because of the deceased family member, even though his or her organs allow new life for other patients. In addition, we thank the FREEZE-biobank-Center for biobanking of the Freiburg University Medical Center and Medical Faculty for support. This work was supported by the project "Virological and immunological determinants of COVID-19 pathogenesis – lessons to get prepared for future pandemics (KA1-Co-02 "COVIPA")", a grant from the Helmholtz Association's Initiative and Networking Fund (to R.T. and M.H.). M.H. is supported by the DFG Heisenberg program (HO-5836/2-1). B.C.F. is supported by the Berta-Ottenstein-Program of the Medical University Center, Faculty of Medicine, University of Freiburg. JF is supported by the Medical-Scientist Program of the Medical University Center, Faculty of Medicine, University of Freiburg.

## Author contributions

J.F. and V.K. planned, performed and analyzed experiments with the help of L.J., A.M., A.-K.K., D.Hu., G.K., A.K., and D.E. I.H., J.A., C.T., P.A., E.S., N. V., G.Z., S.F., D.St., and B.C.F. collected, interpreted and contributed clinical data. I.H., F.E., S.F., A.F., D.Ho., J.K., I.L., A.L., I.M., D.Sc., M. C., B.P., D.St., and B. C. F. were part of the lung transplant team, contributed to data collection and analyses and were involved in the clinical management and the decision for lung transplantation. F.E. performed four-digit HLA-typing by next-generation sequencing. J.F., V.K., N.K., C.N.H., D.St., R.T., M.H., and B.C.F. contributed to data interpretation. M.P., I.H., S.F., D.St., C.N.-H., R.T., and N.K. gave critical intellectual content. J.F., M.P., B.C.F., and M.H. designed and supervised the study. J.F., V.K., M.H., and B.C.F. wrote the manuscript. M.P., M.H., and B.C.F. are shared last authors.

## Funding

## Competing interests

A.L. indicates research grants from the German Research Foundation, the German Heart Foundation and the German Center for Infectious Research. A.L. indicates travel grants from Diaplan and participation in advisory boards of Bayer AG. B.C.F. indicates research support from Bristol-Myer Squibb and Relief Therapeutics unrelated to the manuscript, consulting and lecture fees from Advita Lifescience GmbH, Actelion, AstraZeneca, Boehringer Ingelheim, Novartis, Roche and Vifor, travel support from Boehringer Ingelheim. B.C.F. indicates the following intellectual property: WO2020225246A1; WO2021152119A1. DH indicates lecture fees from BioMerieux. D.St. reports financial support for lectures or participation at advisory boards from Astra-Zeneca AG, Novartis AG, GSK AG, Roche AG, Zambon, Pfizer, Schwabe Pharma AG, Vifor AG, Chiesi AG, MSD, Pfizer, Sanofi, Chemie Menarini, CSL-Behring, Boehringer Ingelheim outside the submitted work. N.K. indicates lecture fees from AstraZeneca. PA indicates personal lecture fees from Boehringer Ingelheim. SF indicates personal lecture fees from Astra Zeneca and CSL Behring outside the submitted work. All other authors declare no competing interests.

## Additional information

[1]Institute of Virology, Freiburg University Medical Center, Faculty of Medicine, University of Freiburg, Freiburg, Germany. [2]Faculty of Biology, University of Freiburg, Freiburg, Germany. [3]Department of Medicine II, Freiburg University Medical Center, Faculty of Medicine, University of Freiburg, Freiburg, Germany. [4]Department of Pneumology, Freiburg University Medical Center, Faculty of Medicine, University of Freiburg, Freiburg, Germany. [5]Department of Medicine V, LMU University Hospital, LMU Munich, Munich, Germany. [6]Department of Radiology, Freiburg University Medical Center, Faculty of Medicine, University of Freiburg, Freiburg, Germany. [7]Institute for Transfusion Medicine and Gene Therapy, Freiburg University Medical Center, Faculty of Medicine, University of Freiburg, Freiburg, Germany. [8]Interdisciplinary Medical Intensive Care, Medical Center - University of Freiburg, Faculty of Medicine, University of Freiburg, Freiburg, Germany. [9]Division of Infectious Diseases, Dept. Med. II, Freiburg University Medical Center, Faculty of Medicine, University of Freiburg, Freiburg, Germany. [10]Department of Anaesthesiology and Critical Care Medicine, Medical Center-University of Freiburg, Faculty of Medicine, University of Freiburg, Freiburg, Germany. [11]Department of Internal Medicine, Division of Respiratory Medicine, Lung Research Cluster, Medical University of Graz, Graz, Austria. [12]Department of Medicine V, LMU University Hospital, LMU Munich, Comprehensive Pneumology Center Munich (CPC-M), Member of the German Center for Lung Research (DZL), 81377 Munich, Germany. [13]Department of Psychosomatic Medicine und Psychotherapy, Center for Mental Health, Faculty of Medicine, University of Freiburg, Freiburg, Germany. [14]Institute of Experimental and Clinical Pharmacology and Toxicology, Faculty of Medicine, University of Freiburg, Freiburg, Germany. [15]Department of Thoracic Surgery, Medical Center-University of Freiburg, Freiburg im Breisgau, Germany. [16]Department of Cardiovascular Surgery, Medical Center University of Freiburg, Faculty of Medicine, University of Freiburg, Freiburg, Germany. [17]Department of Cardiac Surgery, Lucerne Cantonal Hospital, Lucerne, Switzerland. [18]Institute for Surgical Pathology, Medical Center University of Freiburg, Faculty of Medicine, University of Freiburg, Freiburg, Germany. [19]Department of Rheumatology and Clinical Immunology, Freiburg University Medical Center, Faculty of Medicine, University of Freiburg, Freiburg, Germany. [20]Department of Gastroenterology and Hepatology, University Hospital Cologne, Faculty of Medicine, University of Cologne, Cologne, Germany. [21]These authors contributed equally: Jonas Fuchs, Vivien Karl, Ina Hettich. [22]These authors jointly supervised this work: Marcus Panning, Maike Hofmann, Björn C. Frye. ✉e-mail: marcus.panning@uniklinik-freiburg.de; maike.hofmann@uniklinik-freiburg.de; bjoern.christian.frye@uniklinik-freiburg.de

