## [Transparent Peer Review file · Nature Communications]

SARS-CoV-2 infection dynamics in a MHCII-mismatched lung transplant recipient

Corresponding Author: Dr Björn Frye

Version 0:

Reviewer comments:

Reviewer #1

(Remarks to the Author)

In this manuscript Fuchs and colleagues describe case report of a patient who received lung transplantation due to severe COVID-19 infection and underlying interstitial lung disease. The transplanted graft was re-infected leading to virus persistence for more than five months. Patient showed the emergence of variant SARS-CoV-2 which did not react to monoclonal SARS-CoV-2-specific antibodies and hypermutated upon molnupiravir treatment. In spite of strong anti-viral T cell immunity in the transplant recipient, these immune cells did not cross-recognize infected lungs due to HLA mismatch. Based on these observations, authors argue that only donor-derived T-cells can contribute to the antiviral immune response in transplanted organ. Overall this is an interesting case report but I am not sure what is the novelty of the observations described in this manuscript. Authors should specify this clearly. I have few additional comments/suggestions which author may like to consider while revising their manuscript.

(a) Mismatch MHC and its impact on recipient T cell recognition in the engrafted organ is not surprising. Did authors see any evidence of chimerism either in the engrafted organ or peripheral blood.

(b) Did authors HLA class II-restricted CD4 T cells? Virus-specific CD4 T cells can often recognize multiple HLA class II alleles. A more detailed analysis of CD4 T cells is essential to understand the immune regulation of viral infection.

(c) It would be nice if authors can run a deep sequencing TCR analysis of T cells in the lung and peripheral blood. They can use GLIPH software to identify some virus-specific TCRs.

(d) I am not sure if authors have considered trogocytosis where cells from donor can acquire part of the plasma membrane and cytoplasm of recipient cells through direct contact. It will be also interesting to see if there are cells in lungs which have acquired HLA molecules from the recipient which would allow a cross-recognition of virus-infected cells.

(e) I was bit confused while reading the abstract and introduction. In the abstract authors argue the role of mismatched MHC for prolonged infection. However, end of the introduction refers to molnupiravir drive mutagenesis of the viral genome. What is the precise message authors are trying to convey?

Reviewer #2

(Remarks to the Author)

In the present study, Fuchs et al. have made several important findings:

- 1) Lung transplantation can be a viable therapeutic option for patients despite persistence of SARS-COV2, when combined with molnupiravir as post-transplant antiviral regimen.
- 2) In case of indications of SARS-COV2 persistence at least partial HLA matching should be considered to ensure efficient anti-viral control within the lower respiratory tract by recipient T cells.
- 3) Standard diagnostics focus on upper respiratory tract samples to assess an ongoing SARS-CoV2 infection. The present study highlight, importantly, that an ongoing infection (at least in lung transplant patients) can be missed by solely monitoring the viral load within the upper respiratory tract.

The main findings of this study suggest lung transplantation as a treatment for SARS-CoV2 infected patients with severely impaired lung function lacking alternative options. This approach does not align with the current guidelines and should be reviewed in light of the results. The findings encourage further investigation into whether transplanting HLA-matched lungs could improve the chances of success and whether earlier or even preemptive molnupiravir therapy might be advantageous. However, this study is a case-report with a strong focus on the clinical presentation of the patient. There is a lack of mechanistic studies to provide scientific explanations for the described results. Moreover, the clinical section and the findings on viral evolution are tailored to a specialized audience, making it challenging for the broader readership of Nature Communications to fully grasp the narrative.

Major comments and limitations:

- 1) Being a case report the present study does not provide a setting which proves the benefit of a HLA match. This could be only possible in a controlled setting.
- 2) Supplementary Table 1, 2, nor 3 are provided.
- 3) It is not clearly stated how 'response' was assessed in fig. 4d. Should this be the IFN- γ release as shown in fig. 4c, this needs to be clearly stated and a threshold of IFN- γ production should be given (e.g. increase of > 10 % IFN- γ + SARS-COV2-specific CD8+ T cells). Prove of this (e.g. flow cytometry blots, etc. should be provided in the supplements). This applies also to fig. 7a.
- 4) The authors concluded for figure 4e and f that SARS-CoV2-specific recipient T cells did not acquire a tissue-resident memory T (Trm) cell phenotype. This is not correct. The authors firstly draw this conclusion only from the expression of CD103, however, CD103 is one marker, but in human neither the only nor the hallmark marker for Trm cells. This is CD69 (Mueller and Mackay, 2015, Nature Reviews Immunology), which was not investigated or not included in the figures. Therefore, acquisition of a Trm phenotype cannot be concluded from the data presented, the authors can only make speculations via the % of Trm cells expressing CD103 or phrase it as 'Trm cell-characteristics'. Secondly, recipient CD8+ T cells express already at 0m ~15 % CD103, which even increases to about 30 % at 8m. Hence, recipient T cells expressing CD103 double during the investigated time frame, indicating a gain in Trm cell-characteristics.
- 5) Line 344 – The statement that low frequencies of donor T cells were probably insufficient for viral clearance is not supported by the data. There are no indications/reasons provided that there would be lower frequencies of "donor" T cells in the graft than in a non-transplanted lung (which is sufficient to clear ongoing infections).

Minor points and recommendations:

- 1) Line 76 – Stated that 'ARDS-related mortality is high' - please also add reference frame to this subjective statement.
- 2) In general – Inconsistent use of pharmacological and scientific names of drugs (e.g. line 141 tixagevimab/cilgavimab and fig. 1a Evusheld). Please be consistent.
- 3) In general – Explain briefly mode of action of relevant drugs.
- 4) Line 133 – Incorrect referencing of figures. The text refers to fig. 1d, but fig. 1c is referenced.
- 5) Fig 1b – Please indicate (e.g. with arrows) which microanatomical structures are relevant for the reader.
- 6) Line 144 – Unclear use of technical term 'GGO'. Please explain the term GGO, its relevance to the disease pathology and the consequences of finding/ not finding such in the patient.
- 7) Line 149 – Please state whether it can be ruled out, that the patient was (super)infected (i.e. was the patient tested for any other viruses or pathogens).
- 8) Line 170 – Please explain the term 'BA.2', it is not obvious to the non-expert reader.
- 9) Line 175 – Please state whether 10 mutations/ day lies within or deviates from expectations. If possible, consult explicit numbers from literature.
- 10) Line 201 – Please introduce the abbreviation 'RBD'.
- 11) Line 242 – It is known that donor-derived leukocytes can be found in the circulation after transplantation. Please state that this is a possibility and that you did not detect them (Almeida et al., 2022, Science Immunology).
- 12) For ease of comprehension, the figs. 1d, 4a, Suppl. 2a,b should be marked at d0, the day of the transplantation (e.g. with a vertical horizontal line).
- 13) Line 346 – Reference to "18 months" appears to be incorrect. The data only show analyses until 8 months post transplantation.

Clarity of reporting

- 1) Fig 1b + Fig 2 + Suppl. fig. 1b – Not enough explanation provided in order to be easily understandable for non-expert readers. Please guide the reader through the figures and provide explanation for important aspects of each figure.
- 2) Fig. 4b – Please provide information about if or how many technical replicates were included. Please provide more information about the healthy (clearer naming: convalescent) control. Please provide information about whether and if 'yes', why only one healthy control was used. If possible, please include more healthy controls. Also please use consistent time line indications (also fig. 4e,f) – first instance of 'months'.
- 3) Fig. 4b, 4e,f – Please indicate clearly in the figures, that b shows blood T cells and e and f show BAL T cells.
- 4) Suppl. fig. 6 – Please indicate the investigated time point of this example. Also, in future experiments, choosing a different fluorochrome conjugate for every tetramer would be optimal, to be able to assess specificity of the staining by e.g. providing FACS plots pre-gated on CD8+ T cells, showing A*03/S378- tetramer vs. A*03/N361 tetramer. Furthermore, a T cell marker, such as CD3, should be included in the analysis. Please indicate in the figure a note, that the 'blood' samples show specifically enriched cells.
- 5) Line 531 – Please provide a brief description about the magnetic bead enrichment used to identify SARS-CoV2-specific CD8+ T cells in blood samples.

(Remarks to the Author)

In this paper, Fuchs et al evaluated the role of class I MHC complete mismatch on T cell control of SARS-CoV-2 infection in a recipient of lung transplantation (LuTx) for SSc-ILD complicated by COVID-19.

The authors elected to proceed to transplantation despite still detectable SARS-CoV-2, as it was deemed the only therapeutic option available for the patient. After the transplant, the virus did not react to SARS-CoV-2-specific monoclonal antibodies, showed divergent evolution in the recipient and in the donor graft, and hypermutated upon molnupiravir treatment without significant loss in replication capacity, causing a prolonged infection. Only after a second course of molnupiravir viral clearance was reached. Looking at SARS-CoV-2-specific B and T cell immunity, the authors observed the absence of a virus-specific humoral response attributed to the preceding B-cell depleting therapy with rituximab. In contrast, circulating SARS-CoV-2-specific CD8+ T-cells reached frequencies similar to healthy controls. However, diverged viral clearance in the recipient's URT and the donor's LRT was observed, leading to the hypothesis that the different HLA types in the HLA-disparate LuTX may account for the recipient's T-cell failure to control SARS-CoV-2 infection in the transplanted lung. The authors conclude that recipient T-cells in the cognate MHC context may be induced by viral infection, but these cells do not cross-recognize the infected transplanted lungs. In contrast, only donor-derived T-cells can contribute to the antiviral immune response in the allograft.

This is an interesting case that expands knowledge on immune responses to viruses in the context of allotransplantation. The clinical assessment was robustly complemented by viral infection and specific immune response in-depth analysis. In addition, the paper is clearly written and well discussed. The findings derived from the study, although somehow expected, have potential biological and therapeutic interest.

Specific comments:

1. The main limitation of the study lies in the immunological evaluation. Having used tetramer-based flow cytometry technology, the authors limited their analysis to CD8+ T cells. However, CD4+ T cells are known to have a role in SARS-CoV-2 infection control, and they may be important in limiting infection to a MHC-disparate graft, as they are known to be more promiscuous than CD8+ T cells. Therefore, it would have been interesting to also analyze SARS-CoV-2-specific CD4+ T cells, especially after IS tapering.
2. Along the same line, the association of clinical observation and CD8+ T cell analysis by tetramer flow cytometry could have been further corroborated by a functional test (perhaps specific cytokine production in a ICS assay).

Reviewer #4

(Remarks to the Author)

This manuscript by Fuchs et al. details a case study of a lung transplant recipient who was transplanted prior to complete recovery from SARS-CoV-2. The manuscript describes the persistence of virus in the allograft following transplantation. The authors report a distinct viral evolution in the recipient and the allograft, and the virus was hypermutated with molnupiravir treatment. The virus persisted in the lower respiratory tract (BAL), a finding that the authors contributed to a failure of recipient-derived T-cells to recognise virally infected cells in the context of an HLA mismatched allograft.

The strength of the manuscript is the sophisticated virology studies presented, and the evolution of the virus is noteworthy. The manuscript also serves as a potential resource to guide transplant units on how to manage patients with persisting SARS-CoV-2 infection.

The immunological findings of the study are interesting; however, I remain unconvinced that HLA class I mismatching is the cause of viral persistence in the lower respiratory tract. Some reasons for this are as follows:

- Other studies have observed viral persistence in the lower respiratory tract in the absence of an allograft in situ (e.g. Ramos et al., *Respir Med Case Rep.* 2020)
- Adoptive T cell therapy for refractory CMV in lung transplant recipients shows viral clearance from the lung using recipient derived CD8+ T cells (i.e. restricted by allograft-disparate HLA) (e.g. Holmes-Liew et al., *Clin Trans Immunol* 2015)
- A recent study shows T cells play a minimal role in clearing virus from the lower respiratory tract, albeit in mice (Kar et al. *Science Advances*, 2024). If this is translatable to humans, the lack of HLA class I commonality between recipient and donor is irrelevant
- HLA-E restricted, SARS-CoV-2 specific CD8 T cells have been shown to expand to similar levels to classical HLA-I restricted T cells (Yang et al., 2023), and HLA-E, being non-polymorphic, would be equivalent between donor and recipient (i.e. these recipient derived CD8 T cells would cross-react on the allograft)
- Previous studies have shown down-regulated HLA class I in SARS-CoV-2, possibly contributing to viral persistence (e.g. Yoo et al. *Nat. Comms.*, 2021, Zhang et al. *Proc. Nat. Acad. Sc.*, 2021)
- A recent study suggests that tissues may be a reservoir of SARS-CoV-2 and the observations in the study may merely reflect this (Machkovech et al., *Lancet Infectious Diseases*, 2024)

It should also be noted that the recipient/donor HLA class I are a relatively good match, differing only by 16 eplets across HLA-A and -B combined. As such, and combined with the points above, the significance of donor-recipient HLA mismatching is uncertain.

Other questions/comments

- There is no mention of donor or recipient HLA-C or HLA class II, are there overlapping alleles?

- Did the authors investigate NK cells in the blood or lung? Could NK cell depletion/exhaustion be a cause of lack of viral control?
- The authors state that only recipient SARS-CoV-2-specific CD8+ T-cells could be detected in the blood, although BAL samples contained both donor- and recipient-derived T cells. The donor-derived T cells were not detected at follow-up, and the authors therefore concluded that they were derived from a previous donor infection. Although this may be true, there is not enough evidence presented to support this conclusion. Previous studies have shown very low frequency of donor-derived T cells in the blood of transplant recipients, and a higher proportion in the allograft that tends to decrease over time (Snyder et al., Sci. Immunol., 2019). Moreover, one would expect the proportion of donor-derived T cells to be reduced over time, given the reduction of immunosuppression in this patient.
- Page 3 line 85-86, "Additionally, transplantation without prior SARS-CoV-2 clearance increases the risk of SARS-CoV-2 transmission and persistence in the transplanted lung." Is this known? If so, please provide a reference. Further to this, this seems at odds with the statement on page 3 line 89-91 "So far, transmission of SARS-CoV-2 from the recipient to the lung allograft have not been described."
- Page 4 line 1, "lungtransplant" needs a space
- Page 8 line 204. Remove "." At start of paragraph

Reviewer #5

(Remarks to the Author)

Reviewer #6

(Remarks to the Author)

Version 1:

Reviewer comments:

Reviewer #1

(Remarks to the Author)

Authors have addressed all my comments and concerns. They have revised their manuscript and have provided detailed rebuttal. I don't have any further concerns.

Reviewer #2

(Remarks to the Author)

The manuscript has been revised according to the suggestions and has improved in overall quality. However, not all of the changes mentioned in the response letter are reflected in the manuscript, and this should be addressed accordingly.

The main concerns persist, particularly the lack of mechanistic studies and the absence of direct evidence supporting the manuscript's main conclusion—namely, the benefit of HLA matching, as still indicated by the unchanged title.

The authors' point that the manuscript combines clinical application, viral evolution, and basic immunology is valid.

However, the latter two aspects are described rather broadly, without being explored in a mechanistic manner.

Major comments and limitations:

1) Initial comment:

Being a case report the present study does not provide a setting which proves the benefit of a HLA match. This could be only possible in a controlled setting.

The major limitation was acknowledged, comparing HLA-matched versus mismatched transplantation in a controlled setting is indeed not feasible. For this reason, we recommended rephrasing the statements. While the abstract was adjusted accordingly, the title remained unchanged, which still implies a controlled comparison.

2) Supplementary Tables 1, 2 and 3 have now been added and thus the comment was satisfactorily addressed.

3) Initial comment on missing explanation of assessment of T cell response.

The authors have now added a description to the Materials and Methods section. According to this, T-cell responses were determined by measuring intracellular IFN- γ levels, using a cut-off of 0.01%. However, this cut-off appears inconsistent with the data, as the unstimulated control for CD8⁺ T cells already shows 0.033% positive cells in Supplementary Figure 6b.

Additionally, the gating appears to differ between CD4⁺ and CD8⁺ T cells.

Therefore, the strategy for defining a positive response requires revision. We recommend defining a response as any stimulation that results in a higher percentage of positive cells compared to the corresponding unstimulated control.

Moreover, the classification of responses should be clearly indicated in the figure and briefly mentioned in the figure legend.

4) Initial comment concerning the acquisition of a Trm phenotype was satisfactorily addressed.

5) Initial comment concerning the low donor T-cell frequencies was satisfactorily addressed.

Minor points and recommendations:

- 1) Initial comment on ARDS-related mortality was satisfactorily addressed.
- 2) Initial comment on consistent naming was satisfactorily addressed.
- 3) Mode of action for drugs has been added, no further action needed.
- 4) Figures are now correctly referenced, no action needed.
- 5) Although this point was said to be addressed in the response letter, the legend of Figure 1b was not changed, and there are no visible modifications to the figure itself. It remains unclear which anatomical structures are of interest in Figure 1b and Supplementary Figure 1b.
- 6) Was addressed in the text, but the GGOs should be pointed out in the figure.
- 7) Information on tested pathogens was provided, no further action needed.
- 8) Was satisfactorily addressed.
- 9) Was satisfactorily addressed.
- 10) Was satisfactorily addressed.
- 11) Was satisfactorily addressed.
- 12) The authors addressed this point. Our original comment mistakenly read "(e.g. with a vertical horizontal line)". What we meant was not a horizontal, but a vertical line—so that when following the data points, a clear visual separation between the time points before and after transplantation is immediately apparent. We kindly ask the authors to make this adjustment and apologize for the confusion.
- 13) Was satisfactorily addressed.

Clarity of reporting:

- 1) Other than reported by the authors, neither fig. 1 nor suppl. fig 1 legend was changed.
- 2) Was satisfactorily addressed.
- 3) Was satisfactorily addressed.
- 4) Was satisfactorily addressed.
- 5) Was satisfactorily addressed.

Reviewer #3

(Remarks to the Author)

The authors have addressed most of the concerns of the reviewers, including all suggestions from this reviewer, and the paper has greatly gained from the amendments.

Reviewer #4

(Remarks to the Author)

Overall, the manuscript has improved in its clarity and detail of the presented case study. As a case study it does provide insight in how clinical teams can manage COVID infections with patients awaiting lung transplantation, especially in the monitoring and treatment of recent infections. However, I remain unconvinced of the novelty of the findings in a single case study.

Although it makes immunological "sense" there is not enough evidence to conclusively show that that persistent infection in the LRT is due to the disparate donor/recipient HLA and the inability of recipient CD8+ T cells to respond to the infection. For example, the authors now show that only CD4+ T cells of recipient origin can be found in the BAL. Perhaps this is the reason for lack of viral control in the LRT? Moreover, the authors acknowledge the role of NK cells has not been examined, nor the effect of immunosuppression and its effect on donor vs recipient immune cells. Therefore, there could be a multitude of reasons for persistence in the LRT. Although the conclusions have been toned down somewhat, the manuscript title still implies HLA mismatch is the cause of viral persistence. The authors have also not acknowledged that other studies have observed viral persistence in the LRT in non-transplant settings. The fact that recipient CD8+ T cells do not recognize donor HLA in vitro is not a novel finding and may be only part of a far more complex immunological puzzle.

Minor fixes

Page 3 lines 91-93 ** needs referencing "Nevertheless, transmission of SARS-CoV-2 from the recipient to the donor organ is of concern, especially because persistent infection and acquisition of in-host mutations have both been described in organ-transplanted patients."

Page 4 line 101 "to replicatein" needs a space

Page 5 line 134-137 "recipient: HLA-A*03, HLA-B*08, HLA-B*40, 135 HLA-C*03, HLA-C*07, HLA-DRB1*03, HLA-DRB1*04, HLA-DQB1*02, HLA-DQB1*03, 136 HLA-DPB1*01, HLA-DPB1*02 ; donor: HLA-A*01, HLA-A*25, HLA-B*27, HLA-B*37, 137 HLA-C*02, HLA-C*06, HLA-DRB1*01, HLA-DRB1*14, HLA-DQB1*05, HLA-DPB1*04)" Doesn't need "HLA-" before each HLA. Eg. HLA-A*03; B*08; B*40; C*03; C*07..... is sufficient.

Page 6 line 152 "extensive screening.." remove "."

Page 6 line 154 "loadstill" needs space

Page 7 line 172 "antiviral alternative.." remove "."

Page 9 line 226 "OmicronBA.2 variantisolates" needs space

Page 14 line 365 what does "non-polymorphic HLA-based reactivates" mean? This doesn't make sense.

Page 14 line 367 "wether" should be corrected to "whether"

Reviewer #5

(Remarks to the Author)

Reviewer #6

(Remarks to the Author)

Version 2:

Reviewer comments:

Reviewer #2

(Remarks to the Author)

The authors have addressed my concerns sufficiently.

Reviewer #5

(Remarks to the Author)

Point-to-Point Reply

We would like to thank the reviewers in their efforts to improve the manuscript. We feel very appreciated by their time and intellectual input and we have addressed the main issues raised by the reviewers. The major part of the changes (except some typos) are tracked in yellow in the manuscript.

Please find enclosed our detailed comments.

Reviewers' comments:

Reviewer #1 (Remarks to the Author):

In this manuscript Fuchs and colleagues describe case report of a patient who received lung transplantation due to severe COVID-19 infection and underlying interstitial lung disease. The transplanted graft was re-infected leading to virus persistence for more than five months. Patient showed the emergence of variant SARS-CoV-2 which did not react to monoclonal SARS-CoV-2-specific antibodies and hypermutated upon molnupiravir treatment. In spite of strong anti-viral T cell immunity in the transplant recipient, these immune cells did not cross-recognize infected lungs due to HLA mismatch. Based on these observations, authors argue that only donor-derived T-cells can contribute to the antiviral immune response in transplanted organ. Overall, this is an interesting case report but I am not sure what is the novelty of the observations described in this manuscript. Authors should specify this clearly. I have few additional comments/suggestions which author may like to consider while revising their manuscript.

We would like to thank the reviewer for the helpful comments, especially for pointing out that the novelty of our observations is not properly presented. Following the suggestion, in the revised manuscript we have clarified our three main findings:

- a) Viral evolution and clearance may diverge between upper (HLA-synonymous) and lower (HLA-disparate) respiratory tract. This result questions current monitoring strategies with implications for prolonged viral persistence in HLA-mismatch lung transplant recipients.*
- b) In a patient in vivo, viral clearance depends on a compatible immune system with differential contribution of the donor and recipient T cells to immunosurveillance of a transplanted organ. To the best of our knowledge this has not been demonstrated in a clinical setting of lung transplantation and therefore shows that basic immunological principles are relevant in the clinical practise and may impact outcome after lung transplantation.*
- c) Lung transplantation is feasible despite non-complete viral clearance challenging current guidelines.*

(a) Mismatch MHC and its impact on recipient T cell recognition in the engrafted organ is not surprising. Did authors see any evidence of chimerism either in the engrafted organ or peripheral blood.

We find a T cell chimerism in the BAL (reflecting the alveolar compartment of the lung transplant) that declines with time (Figure 4e+f and Supplementary Figure 8). T cell chimerism was not observed within the blood (Figure 4c and Supplementary Figure 6b). These observations are in accordance with other reports by Snyder et al., Sci Immunol 2019 and Bellmàs-Sanz et al., bioRxiv 2023. We have now more clearly labelled data of T cells obtained from BAL or blood in figure 4.

(b) Did authors HLA class II-restricted CD4 T cells? Virus-specific CD4 T cells can often recognize multiple HLA class II alleles. A more detailed analysis of CD4 T cells is essential to understand the immune regulation of viral infection.

*We would like to thank the reviewer for this valuable suggestion. We analyzed CD4+ T cell immunity and indeed detected SARS-CoV-2-reactive CD4+ T cells in the blood of the patient. In addition to SARS-CoV-2-reactive CD4+ T cells towards recipient-specific epitopes we also found CD4+ T cells that were reactive towards SARS-CoV-2-derived epitopes restricted by multiple HLA class II alleles so-called promiscuous epitopes and can consequently be recognized by CD4+ T cells from the donor and the recipient. However, we only recovered CD4+ T cells from the patient (recipient, HLA-A*03+) and not from the donor in the blood and BAL. We have included this data set in Supplementary Figure 6a+c of the revised manuscript. Additionally, we now also show that SARS-CoV-2-specific CD4+ T cell epitopes were conserved in the virus isolates of the patient suggesting limited immune selection pressure (Supplementary Figure 9b).*

(c) It would be nice if authors can run a deep sequencing TCR analysis of T cells in the lung and peripheral blood. They can use GLIPH software to identify some virus-specific TCRs.

We would like to thank the reviewer for this suggestion. We absolutely agree that with this approach we would also be able to detect several epitope-specific T cell populations. Unfortunately, we do not have enough sample material left to perform deep TCR sequencing.

(d) I am not sure if authors have considered trogocytosis where cells from donor can acquire part of the plasma membrane and cytoplasm of recipient cells through direct contact. It will be also interesting to see if there are cells in lungs which have acquired HLA molecules from the recipient which would allow a cross-recognition of virus-infected cells.

We cannot completely rule out a role of trogocytosis in the presented case, but cross-recognition by trogocytosis appears not to play a major role in our patient since SARS-CoV-2 was not cleared in a short timeframe after reduction of the immunosuppressive therapy. To further follow up on this suggestion, we, unfortunately, do not have enough sample material left.

(e) I was bit confused while reading the abstract and introduction. In the abstract authors argue the role of mismatched MHC for prolonged infection. However, end of the introduction refers to molnupiravir drive mutagenesis of the viral genome. What is the precise message authors are trying to convey?

We would like to thank the reviewer for pointing out that the abstract and introduction require revision. We have rephrased the abstract and introduction to more clearly convey the following points:

- a) Viral evolution and clearance may diverge between upper (HLA-synonymous) and lower (HLA-disparate) respiratory tract. This result questions current monitoring strategies with implications for prolonged viral persistence in HLA-mismatch lung transplant recipients.*
- b) Clinical outcome of lung transplantation may be affected by viral clearance that depends on a compatible immune system with differential contribution of the donor and recipient T cells to immunosurveillance of a transplanted organ.*
- c) Lung transplantation is feasible despite non-complete viral clearance.*

Reviewer #2 (Remarks to the Author):

In the present study, Fuchs et al. have made several important findings:

- 1) Lung transplantation can be a viable therapeutic option for patients despite persistence of SARS-CoV2, when combined with molnupiravir as post-transplant antiviral regimen.
- 2) In case of indications of SARS-CoV2 persistence at least partial HLA matching should be considered to ensure efficient anti-viral control within the lower respiratory tract by recipient T cells.
- 3) Standard diagnostics focus on upper respiratory tract samples to assess an ongoing SARS-CoV2 infection. The present study highlight, importantly, that an ongoing infection (at least in lung transplant patients) can be missed by solely monitoring the viral load within the upper respiratory tract.

The main findings of this study suggest lung transplantation as a treatment for SARS-CoV2 infected patients with severely impaired lung function lacking alternative options. This approach does not align with the current guidelines and should be reviewed in light of the results. The findings encourage further investigation into whether transplanting HLA-matched lungs could improve the chances of success and whether earlier or even preemptive molnupiravir therapy might be advantageous.

However, this study is a case-report with a strong focus on the clinical presentation of the patient. There is a lack of mechanistic studies to provide scientific explanations for the described results. Moreover, the clinical section and the findings on viral evolution are tailored to a specialized audience, making it challenging for the broader readership of Nature Communications to fully grasp the narrative.

We would like to thank the reviewer for the valuable comments. In our view, the presented case report combines clinical application, viral evolution and basic immunology and demonstrates that all three aspects converge in translation to the clinic. Based on this we are convinced that the manuscript is of interest for a broad readership of basic and translational scientists as well as physicians. We rephrased the manuscript to make it easier for the reader to follow.

Major comments and limitations:

- 1) Being a case report the present study does not provide a setting which proves the benefit of a HLA match. This could be only possible in a controlled setting.

We completely agree with the reviewer that absolute proof of the HLA-match benefit would require a controlled setting. However, a controlled setting comparing HLA-matched versus -mismatched transplantation is not feasible due to shortage of organ donation. In the revised manuscript, we discuss this limitation.

2) Supplementary Table 1, 2, nor 3 are provided.

We would like to apologize for this inconvenience. The Supplementary Tables are now provided.

3) It is not clearly stated how 'response' was assessed in fig. 4d. Should this be the IFN- γ release as shown in fig. 4c, this needs to be clearly stated and a threshold of IFN- γ production should be given (e.g. increase of > 10 % IFN- γ + SARS-COV2-specific CD8+ T cells). Prove of this (e.g. flow cytometry blots, etc. should be provided in the supplements). This applies also to fig. 7a.

We would like to thank the reviewer for pointing this out. We have now clarified the definition of "T cell response" in the revised Materials and Methods section: "T-cell responses were determined by measuring IFN- γ production and using a cut-off of 0.01 %." In addition, we have also included representative plots depicting IFN- γ production of T cells with the respective controls in Supplementary Figure 6a+b.

4) The authors concluded for figure 4e and f that SARS-CoV2-specific recipient T cells did not acquire a tissue-resident memory T (Trm) cell phenotype. This is not correct. The authors firstly draw this conclusion only from the expression of CD103, however, CD103 is one marker, but in human neither the only nor the hallmark marker for Trm cells. This is CD69 (Mueller and Mackay, 2015, Nature Reviews Immunology), which was not investigated or not included in the figures. Therefore, acquisition of a Trm phenotype cannot be concluded from the data presented, the authors can only make speculations via the % of Trm cells expressing CD103 or phrase it as 'Trm cell-characteristics'.

Secondly, recipient CD8+ T cells express already at 0m ~15 % CD103, which even increases to about 30 % at 8m. Hence, recipient T cells expressing CD103 double during the investigated time frame, indicating a gain in Trm cell-characteristics.

We would like to apologize for this mistake and, of course, we completely agree with the reviewer that indeed the cells acquire a tissue-resident memory phenotype. We corrected the respective sentence and also extended this point by showing CD69/CD103 (Figure 4f), CD69 and CD103 (Supplementary Figure 8) expression on SARS-CoV-2-specific CD8+ T cells.

5) Line 344 – The statement that low frequencies of donor T cells were probably insufficient for viral clearance is not supported by the data. There are no indications/reasons provided that there would be lower frequencies of "donor" T cells in the graft than in a non-transplanted lung (which is sufficient to clear ongoing infections).

We agree with the reviewer that we do not provide corresponding T cell frequencies in non-transplanted lungs. Since we do not have BAL samples from otherwise healthy SARS-CoV-2-infected -or convalescent individuals we toned down this statement.

Minor points and recommendations:

1) Line 76 – Stated that ‘ARDS-related mortality is high’ - please also add reference frame to this subjective statement.

We have included percentages of ARDS-related mortality (25-46%) and respective reference in the revised introduction.

2) In general – Inconsistent use of pharmacological and scientific names of drugs (e.g. line 141 tixagevimab/cilgavimab and fig. 1a Evusheld).

Please be consistent.

Thank you for pointing this out, we now consistently use active substances, e.g. tixagevimab/cilgavimab

3) In general – Explain briefly mode of action of relevant drugs.

We have now included this information in revised Supplementary Table 1.

4) Line 133 – Incorrect referencing of figures. The text refers to fig. 1d, but fig. 1c is referenced.

We have corrected this mistake in the revised manuscript.

5) Fig 1b – Please indicate (e.g. with arrows) which microanatomical structures are relevant for the reader.

We have clarified this point in the revised legend to Figure 1b.

6) Line 144 – Unclear use of technical term ‘GGO’. Please explain the term GGO, its relevance to the disease pathology and the consequences of finding/ not finding such in the patient.

We have rephrased this paragraph to better explain the ground-glass opacity (GGO) as general sign of infection, interstitial lung disease or pulmonary edema and have added an additional reference.

7) Line 149 – Please state whether it can be ruled out, that the patient was (super)infected (i.e. was the patient tested for any other viruses or pathogens).

We tested several viral, bacterial and fungal infections and could not detect any of these tested pathogens. We have included this information in the revised Results section.

8) Line 170 – Please explain the term ‘BA.2’, it is not obvious to the non-expert reader.

We have further clarified BA.2 as SARS-CoV-2 Omicron variant in the revised manuscript.

9) Line 175 – Please state whether 10 mutations/ day lies within or deviates from expectations. If possible, consult explicit numbers from literature.

The calculated mutation rate of 1 mutation per 10 days fits to the reported mutation rates in the general population (Abbasian et al., J Transl Med 2023 and Amicone et al., Evol Med Public Health 2022). We have included these references into the revised manuscript. Furthermore, we rephrased this paragraph to more clearly convey that we have constant acquisition of mutations with an increased mutation rate observed after treatment initiation with molnupiravir.

10) Line 201 – Please introduce the abbreviation ‘RBD’.

We have included the abbreviation RBD – “receptor binding domain”- into the revised manuscript.

11) Line 242 – It is known that donor-derived leukocytes can be found in the circulation after transplantation. Please state that this is a possibility and that you did not detect them (Almeida et al., 2022, Science Immunology).

We would like to thank the reviewer for this comment to extend on the T cell chimerism after transplantation. In the revised manuscript, we further discuss this point, e.g. by referring to a preprint of Bellmàs-Sanz et al., bioRxiv in 2023 showing that donor T cells persist in the circulation 3 weeks after lung transplantation.

12) For ease of comprehension, the figs. 1d, 4a, Suppl. 2a,b should be marked at d0, the day of the transplantation (e.g. with a vertical horizontal line).

We have adapted the figures by shading the respective time-frame after the day of transplantation in light grey.

13) Line 346 – Reference to “18 months” appears to be incorrect. The data only show analyses until 8 months post transplantation.

We have corrected this mistake in the revised manuscript.

Clarity of reporting

1) Fig 1b + Fig 2 + Suppl. fig. 1b – Not enough explanation provided in order to be easily understandable for non-expert readers. Please guide the reader through the figures and provide explanation for important aspects of each figure.

In the revised figures and corresponding legend, we provide additional information for a better guidance of the reader.

2) Fig. 4b – Please provide information about if or how many technical replicates were included.

Please provide more information about the healthy (clearer naming: convalescent) control. Please provide information about whether and if 'yes', why only one healthy control was used. If possible, please include more healthy controls. Also please use consistent time line indications (also fig. 4e,f) – first instance of 'months'.

We would like to thank the reviewer for this comment. We have included additional details in the figure legend. Furthermore, we have added more data points on SARS-CoV-2 convalescent donors and rearranged the figures, e.g. using consistent time-lines.

3) Fig. 4b, 4e,f – Please indicate clearly in the figures, that b shows blood T cells and e and f show BAL T cells.

We would like to thank the reviewer for this suggestion and indicate blood- and BAL-derived T cells, accordingly.

4) Suppl. fig. 6 – Please indicate the investigated time point of this example. Also, in future experiments, choosing a different fluorochrome conjugate for every tetramer would be optimal, to be able to assess specificity of the staining by e.g. providing FACS plots pre-gated on CD8+ T cells, showing A*03/S378- tetramer vs. A*03/N361 tetramer. Furthermore, a T cell marker, such as CD3, should be included in the analysis.

Please indicate in the figure a note, that the 'blood' samples show specifically enriched cells.

We would like to thank the reviewer for these valuable suggestions. In the revised manuscript, we have included the investigated time point of the example in the figure legend. Moreover, we have better clarified that the SARS-CoV-2-specific CD8+ T cells were enriched based on pMHC class I tetramers - magnetic beads. A detailed gating strategy is provided in Supplementary Figure 7. CD3 was not included in the panel due to parameter restrictions.

5) Line 531 – Please provide a brief description about the magnetic bead enrichment used to identify SARS-CoV2-specific CD8+ T cells in blood samples.

We followed this suggestion by reviewer #2 and extended the paragraph in the Materials and Methods section.

Reviewer #3 (Remarks to the Author):

In this paper, Fuchs et al evaluated the role of class I MHC complete mismatch on T cell control of SARS-CoV-2 infection in a recipient of lung transplantation (LuTx) for SSc-ILD complicated by COVID-19.

The authors elected to proceed to transplantation despite still detectable SARS-CoV-2, as it was deemed the only therapeutic option available for the patient. After the transplant, the virus did not react to SARS-CoV-2-specific monoclonal antibodies, showed divergent evolution in the recipient and in the donor graft, and hypermutated upon molnupiravir treatment without significant loss in

replication capacity, causing a prolonged infection. Only after a second course of molnupiravir viral clearance was reached. Looking at SARS-CoV-2-specific B and T cell immunity, the authors observed the absence of a virus-specific humoral response attributed to the preceding B-cell depleting therapy with rituximab. In contrast, circulating SARS-CoV-2-specific CD8+ T-cells reached frequencies similar to healthy controls. However, diverged viral clearance in the recipient's URT and the donor's LRT was observed, leading to the hypothesis that the different HLA types in the HLA-disparate LuTX may account for the recipient's T-cell failure to control SARS-CoV-2 infection in the transplanted lung. The authors conclude that recipient T-cells in the cognate MHC context may be induced by viral infection, but these cells do not cross-recognize the infected transplanted lungs. In contrast, only donor-derived T-cells can contribute to the antiviral immune response in the allograft.

This is an interesting case that expands knowledge on immune responses to viruses in the context of allotransplantation. The clinical assessment was robustly complemented by viral infection and specific immune response in-depth analysis. In addition, the paper is clearly written and well discussed. The findings derived from the study, although somehow expected, have potential biological and therapeutic interest.

We would like to thank the reviewer for the positive feedback on our study highlighting the interest for basic and translational scientists. We have addressed the valuable comments as specified below.

Specific comments:

1. The main limitation of the study lies in the immunological evaluation. Having used tetramer-based flow cytometry technology, the authors limited their analysis to CD8+ T cells. However, CD4+ T cells are known to have a role in SARS-CoV-2 infection control, and they may be important in limiting infection to a MHC-disparate graft, as they are known to be more promiscuous than CD8+ T cells. Therefore, it would have been interesting to also analyze SARS-CoV-2-specific CD4+ T cells, especially after IS tapering.

We would like to thank the reviewer for this highly appreciated suggestion. In the revised version of the manuscript, we have included data showing SARS-CoV-2-specific CD4+ T cell responses in the circulation when immunosuppressants were applied at maintenance level (Supplementary Figure 6a). We detected CD4+ T cell responses that are reactive to recipient-specific epitopes and also directed towards promiscuous epitopes that can be recognized by differentially restricted CD4+ T cells including donor CD4+ T cells. Furthermore, we also observed that SARS-CoV-2-specific CD4+ T cell epitopes were conserved in the virus isolates of the patient suggesting limited immune selection pressure (Supplementary Figure 9b).

2. Along the same line, the association of clinical observation and CD8+ T cell analysis by tetramer flow cytometry could have been further corroborated by a functional test (perhaps specific cytokine production in a ICS assay).

We would like to thank the reviewer for this valuable comment. We have now included data on cytokine production (IFN-g, TNF) and degranulation (CD107a) by SARS-CoV-2-specific CD8+ T cells in revised Figure 4d. SARS-CoV-2-specific CD8+ T cells from the recipient showed robust functional capacities.

Reviewer #4 (Remarks to the Author):

This manuscript by Fuchs et al. details a case study of a lung transplant recipient who was transplanted prior to complete recovery from SARS-CoV-2. The manuscript describes the persistence of virus in the allograft following transplantation. The authors report a distinct viral evolution in the recipient and the allograft, and the virus was hypermutated with molnupiravir treatment. The virus persisted in the lower respiratory tract (BAL), a finding that the authors contributed to a failure of recipient-derived T-cells to recognise virally infected cells in the context of an HLA mismatched allograft.

The strength of the manuscript is the sophisticated virology studies presented, and the evolution of the virus is noteworthy. The manuscript also serves as a potential resource to guide transplant units on how to manage patients with persisting SARS-CoV-2 infection.

We would like to thank the reviewer for acknowledging the impact of our study on our understanding of viral evolution and patient management in transplant units. Furthermore, we would also express our gratitude for the valuable suggestions to improve the immunological part of our study.

The immunological findings of the study are interesting; however, I remain unconvinced that HLA class I mismatching is the cause of viral persistence in the lower respiratory tract. Some reasons for this are as follows:

- Other studies have observed viral persistence in the lower respiratory tract in the absence of an allograft in situ (e.g. Ramos et al., Respir Med Case Rep. 2020)
- Adoptive T cell therapy for refractory CMV in lung transplant recipients shows viral clearance from the lung using recipient derived CD8+ T cells (i.e. restricted by allograft-disparate HLA) (e.g. Holmes-Liew et al., Clin Trans Immunol 2015)
- A recent study shows T cells play a minimal role in clearing virus from the lower respiratory tract, albeit in mice (Kar et al. Science Advances, 2024). If this is translatable to humans, the lack of HLA class I commonality between recipient and donor is irrelevant
- HLA-E restricted, SARS-CoV-2 specific CD8 T cells have been shown to expand to similar levels to classical HLA-I restricted T cells (Yang et al., 2023), and HLA-E, being non-polymorphic, would be equivalent between donor and recipient (i.e. these recipient derived CD8 T cells would cross-react on the allograft)

- Previous studies have shown down-regulated HLA class I in SARS-CoV-2, possibly contributing to viral persistence (e.g. Yoo et al. Nat. Comms., 2021, Zhang et al. Proc. Nat. Acad. Sc., 2021)
- A recent study suggests that tissues may be a reservoir of SARS-CoV-2 and the observations in the study may merely reflect this (Machkovech et al., Lancet Infectious Diseases, 2024) It should also be noted that the recipient/donor HLA class I are a relatively good match, differing only by 16 eplets across HLA-A and -B combined. As such, and combined with the points above, the significance of donor-recipient HLA mismatching is uncertain.

We thank the reviewer for the immunological input concerning the HLA class I mismatch. We agree that not only HLA class I mismatch can result in viral persistence but rather immunosuppression as already shown in several publications (Weigang et al. Nat Communications 2021; Jaki et al., Nature communications 2023, Glueck et al., Infection 2024). However, in this manuscript, we demonstrate clearly, that the immunological chimerism observed in the alveolar compartment of the lung transplant has functional implications, as de novo infiltrating recipient CD8+ T cells could not recognize SARS-CoV-2 epitopes in the HLA class I-disparate organ. This implies that they most likely do not contribute to viral clearance, even though they have antiviral capacity within the blood. We cannot rule out that for example HLA-E-restricted SARS-CoV-2-specific T cells contribute to viral clearance. We have addressed these issues in more detail in the discussion section.

Other questions/comments

- There is no mention of donor or recipient HLA-C or HLA class II, are there overlapping alleles?

We apologize for not providing this information in the initial manuscript, we have added them in the revised version of the manuscript.

- Did the authors investigate NK cells in the blood or lung? Could NK cell depletion/exhaustion be a cause of lack of viral control?

We restricted our analysis to the T cell compartment because of HLA mismatch transplant constellation and limited biomaterials. We can therefore not rule out that NK cells contribute to lacking viral control.

- The authors state that only recipient SARS-CoV-2-specific CD8+ T-cells could be detected in the blood, although BAL samples contained both donor- and recipient-derived T cells. The donor-derived T cells were not detected at follow-up, and the authors therefore concluded that they were derived from a previous donor infection. Although this may be true, there is not enough evidence presented to support this conclusion. Previous studies have shown very low frequency of donor-derived T cells in the blood of transplant recipients, and a higher proportion in the allograft that tends to decrease over time (Snyder et al., Sci. Immunol., 2019). Moreover, one would expect the proportion of donor-derived T cells to be reduced over time, given the reduction of immunosuppression in this patient.

We thank the reviewer for highlighting this point as it was unclear in the manuscript. We have clarified this aspect in the revised version of the manuscript and the corresponding figure to emphasize that our data are in line with the study from Snyder et al., Sci Immunol 2019 and Bellmàs-Sanz et al., bioRxiv 2023.

- Page 3 line 85-86, “Additionally, transplantation without prior SARS-CoV-2 clearance increases the risk of SARS-CoV-2 transmission and persistence in the transplanted lung.” Is this known? If so, please provide a reference. Further to this, this seems at odds with the statement on page 3 line 89-91 “So far, transmission of SARS-CoV-2 from the recipient to the lung allograft have not been described.”

We removed this misleading phrasing. The key message that the recipient should not have signs of overt respiratory tract infections and the resulting recommendations of the ISHLT (https://www.ishlt.org/docs/default-source/uploadedfiles/documents/sars-cov-2-guidance-for-cardiothoracic-transplant-and-vad-center.pdf?sfvrsn=141775db_0) are still mentioned.

- Page 4 line 1, “lungtransplant” needs a space

We have corrected this typo.

- Page 8 line 204. Remove “.” At start of paragraph

We have corrected this typo.

Reviewer #5 (Remarks to the Author):

We would like to thank the Early Career Researcher for the valuable comments that helped to revise and improve our manuscript.

Reviewer #6 (Remarks to the Author):

We would like to thank the Early Career Researcher for the helpful comments to revise and our manuscript. Through revision we are convinced that our manuscript has improved.

Point-to-Point Reply

“Deciphering the *in vivo* relevance of MHC mismatch for SARS-CoV-2 infection in lung transplantation”

New title: “SARS-CoV-2 infection dynamics in a MHCI-mismatched lung transplant recipient”

Reviewers' comments:

Reviewer #1 (Remarks to the Author):

Authors have addressed all my comments and concerns. They have revised their manuscript and have provided detailed rebuttal. I don't have any further concerns.

We would like to thank the reviewer for taking the time to review our manuscript and the helpful comments.

Reviewer #2 (Remarks to the Author):

The manuscript has been revised according to the suggestions and has improved in overall quality. However, not all of the changes mentioned in the response letter are reflected in the manuscript, and this should be addressed accordingly. The main concerns persist, particularly the lack of mechanistic studies and the absence of direct evidence supporting the manuscript's main conclusion—namely, the benefit of HLA matching, as still indicated by the unchanged title. The authors' point that the manuscript combines clinical application, viral evolution, and basic immunology is valid. However, the latter two aspects are described rather broadly, without being explored in a mechanistic manner.

We would like to thank the reviewer for acknowledging the changes made in the revised manuscript. To further improve our manuscript, we have revised our manuscript according to the suggestions made by the reviewer and as detailed below. In particular, we have changed the title.

Major comments and limitations:

1) Initial comment:

Being a case report the present study does not provide a setting which proves the benefit of a HLA match. This could be only possible in a controlled setting. The major limitation was acknowledged, comparing HLA-matched versus mismatched transplantation in a controlled setting is indeed not feasible. For this reason, we recommended rephrasing the statements. While the abstract was adjusted accordingly, the title remained unchanged, which still implies a controlled comparison.

Following the suggestion of the reviewer we have changed the title to “SARS-CoV-2 infection dynamics in a MHCI-mismatched lung transplant recipient”.

2) Initial comment on missing explanation of assessment of T cell response. The authors have now added a description to the Materials and Methods section. According to this, T-cell responses were determined by measuring intracellular IFN- γ levels, using a cut-off of 0.01%. However, this cut-off appears inconsistent with the data, as the unstimulated control for CD8⁺ T cells already shows 0.033% positive cells in Supplementary Figure 6b. Additionally, the gating appears to differ between CD4⁺ and CD8⁺ T cells. Therefore, the strategy for defining a positive response requires revision.

We recommend defining a response as any stimulation that results in a higher percentage of positive cells compared to the corresponding unstimulated control. Moreover, the classification of responses should be clearly indicated in the figure and briefly mentioned in the figure legend.

We would like to thank the reviewer for pointing this out. We have now clarified the definition of “T cell response” in the revised Materials and Methods section (line 581ff): “T-cell responses were determined by subtracting the signal detected in unstimulated samples from stimulated samples and subsequently applying a cut-off of 0.01 %.” The gating strategy for CD4+ and CD8+ T cells varied depending on the separation between positive and negative populations, as well as the expression levels of the markers and background signals in the unstimulated controls (Point-to-point Fig.1). To ensure consistency, gating was always defined based on the corresponding unstimulated (negative control) samples and subsequently applied to the peptide-stimulated samples. This approach, which was used for determining both CD4+ and CD8+ T cell responses, has been applied and validated in different studies on antigen-specific T cells (Lang-Meli et al., Nat Microbiol 2022, Oberhardt et al., Nature 2021, Luxenburger et al., J Hepatol 2023).

Point to point Fig.1: Gating strategy to determine IFN γ response by CD4+ and CD8+ T cells.

We have included additional information on how the T cell response was determined in the revised legend to Supplemental Figure 6a+b.

Minor points and recommendations:

- 1) Although this point was said to be addressed in the response letter, the legend of Figure 1b was not changed, and there are no visible modifications to the figure itself. It remains unclear which anatomical structures are of interest in Figure 1b and Supplementary Figure 1b.

Following the suggestion of the reviewer we have now added arrows (indicate subtle reticulations corresponding to interstitial lung diseases) and asterisks (denoting GGOs and consolidations as surrogates of inflammation) clarifying structures of interest within the pictures of Fig. 1b and Supplementary Figure 1b. Figure 1b additionally depicts a schematic presentation with indicated planes to facilitate orientation.

- 2) Was addressed in the text, but the GGOs should be pointed out in the figure.

According to this suggestion, the GGOs are now pointed out in the figure.

- 3) The authors addressed this point. Our original comment mistakenly read “(e.g. with a vertical horizontal line)”. What we meant was not a horizontal, but a vertical line—so that when following the data points, a clear visual separation between the time points before and after transplantation is immediately apparent. We kindly ask the authors to make this adjustment and apologize for the confusion.

We have now included a vertical line.

Clarity of reporting:

- 1) Other than reported by the authors, neither fig. 1 nor suppl. fig 1 legend was changed.

We have changed the Fig.1 and Suppl. Fig. 1 legends according to the changes made during the revision process.

Reviewer #3 (Remarks to the Author):

The authors have addressed most of the concerns of the reviewers, including all suggestions from this reviewer, and the paper has greatly gained from the amendments.

We would like to thank the reviewer the helpful comments that helped to improve our manuscript.

Reviewer #4 (Remarks to the Author):

Overall, the manuscript has improved in its clarity and detail of the presented case study. As a case study it does provide insight in how clinical teams can manage COVID infections with patients awaiting lung transplantation, especially in the monitoring and treatment of recent infections. However, I remain unconvinced of the novelty of the findings in a single case study.

Although it makes immunological “sense” there is not enough evidence to conclusively show that that persistent infection in the LRT is due to the disparate donor/recipient HLA and the inability of recipient CD8+ T cells to respond to the infection. For example, the authors now show that only CD4+ T cells of recipient origin can be found in the BAL. Perhaps this is the reason for lack of viral control in the LRT? Moreover, the authors acknowledge the role of NK cells has not been examined, nor the effect of immunosuppression and its effect on donor vs recipient immune cells. Therefore, there could be a multitude of reasons for persistence in the LRT. Although the conclusions have been toned down somewhat, the manuscript title still implies HLA mismatch is the cause of viral persistence. The authors have also not acknowledged that other studies have observed viral persistence in the LRT in non-transplant settings. The fact that recipient CD8+ T cells do not recognize donor HLA in vitro is not a novel finding and may be only part of a far more complex immunological puzzle.

We would like to thank the reviewer for acknowledging the insight that has been gained through studying the presented case. As suggested, we have now changed the title to further tone down the conclusion. In addition, we now better acknowledge the probably more complex immunological puzzle associated with viral persistence.

Minor fixes:

Page 3 lines 91-93 ** needs referencing “Nevertheless, transmission of SARS-CoV-2 from the recipient to the donor organ is of concern, especially because persistent infection and acquisition of in-host mutations have both been described in organ-transplanted patients.”

We have included the following references: Weigang et al., Nat Commun 2021 and Jaki et al., Nat Commun 2023.

Page 4 line 101 “to replicatein” needs a space

We have corrected this typo.

Page 5 line 134-137 “recipient: HLA-A*03, HLA-B*08, HLA-B*40, 135 HLA-C*03, HLA-C*07, HLA-DRB1*03, HLA-DRB1*04, HLA-DQB1*02, HLA-DQB1*03, 136 HLA-DPB1*01, HLA-DPB1*02 ; donor: HLA-A*01, HLA-A*25, HLA-B*27, HLA-B*37, 137 HLA-C*02, HLA-C*06, HLA-DRB1*01, HLADRB1*14, HLA-DQB1*05, HLA-DPB1*04)” Doesn’t need “HLA-“ before each HLA. Eg. HLA-A*03; B*08; B*40; C*03; C*07..... is sufficient.

We have changed it, accordingly.

Page 6 line 152 “extensive screening..” remove “.”

We have removed it, accordingly.

Page 6 line 154 “loadstill” needs space

We have corrected this typo.

Page 7 line 172 “antiviral alternative..” remove “.’

We have removed it, accordingly.

Page 9 line 226 “OmicronBA.2 variantisolates” needs space

We have corrected this typo.

Page 14 line 365 what does “non-polymorphic HLA-based reactivates” mean? This doesn’t make sense.

We would like to apologize for the confusion. We have rephrased it to “non-polymorphic HLA-E”.

Page 14 line 367 “wether” should be corrected to “whether”

We have corrected this typo.